# Comparative Fitting of Mathematical Models to Carvedilol Release Profiles Obtained from Hypromellose Matrix Tablets

**DOI:** 10.3390/pharmaceutics16040498

**Published:** 2024-04-04

**Authors:** Tadej Ojsteršek, Franc Vrečer, Grega Hudovornik

**Affiliations:** 1KRKA, d. d., 8501 Novo Mesto, Slovenia; 2Faculty of Pharmacy, University of Ljubljana, 1000 Ljubljana, Slovenia

**Keywords:** hypromellose, HPMC, controlled release, modified release, extended release, prolonged release, modelling, mathematical, drug release

## Abstract

The mathematical models available in DDSolver were applied to experimental dissolution data obtained by analysing carvedilol release from hypromellose (HPMC)-based matrix tablets. Different carvedilol release profiles were generated by varying a comprehensive selection of fillers and carvedilol release modifiers in the formulation. Model fitting was conducted for the entire relevant dissolution data, as determined by using a paired *t*-test, and independently for dissolution data up to approximately 60% of carvedilol released. The best models were selected based on the residual sum of squares (RSS) results used as a general measure of goodness of fit, along with the utilization of various criteria for visual assessment of model fit and determination of the acceptability of estimated model parameters indicating burst release or lag time concerning experimental dissolution results and previous research. In addition, a model-dependent analysis of carvedilol release mechanisms was carried out.

## 1. Introduction

The fitting of mathematical models to drug release data is an important tool in the research and development of drug delivery systems (DDSs). This approach was developed hand in hand with the controlled-release (CR) DDSs and helped the field mature. Mathematical models of drug release, after being fitted to experimental dissolution data, predict drug release as a function of time. They are mostly empirical or semi-empirical and are more often used than comprehensive mechanistic models or statistical models. Over the years, numerous mathematical models were developed, which describe and explain the release of active pharmaceutical ingredients (APIs) from DDSs. The most important are the Zero–order model, the First–order model, the Higuchi model, the Korsmeyer–Peppas model, and the Hixson–Crowell model; however, the Hopfenberg model, the Baker–Lonsdale model, the Peppas–Sahlin model, the Weibull model, and others are sometimes used as well. Some of these models can be thought of as mathematical metaphors of actual physical phenomena ruling drug release kinetics and therefore have some rooting in reality, although they are not comprehensive mechanistic models. Models which fall into this category are, for example, the Higuchi model, the Korsmeyer–Peppas model, the Peppas–Sahlin model, the Hixson–Crowell model, the Hopfenberg model, the Baker–Lonsdale model, and the Makoid–Banakar model. In contrast, the Zero–order model simply describes the API release at a constant rate as a function of time and independent of the API concentration [1]. It can also be thought of as a variation of the Korsmeyer–Peppas model along with the Higuchi model. The First–order model states that the API release rate is dependent only on the API concentration. Other models, like the Quadratic model, the Logistic model, the Gompertz model, and the Probit model, do not have any rooting in the reality of drug release phenomena. They represent flexible mathematical functions, which were adapted for fitting to drug release data from other fields, as they proved useful in modelling drug release [1,2,3,4,5,6,7,8,9,10]. The Weibull model was once considered as part of this latter group [1,2,4]. However, despite being an empirical model, recent studies have shown that it has some basis in the reality of drug release phenomena. In fact, it can even be utilized in model-dependent analyses of drug release mechanisms [11,12].

Mathematical models, after being fitted to experimental dissolution data, have several main applications:-Representation of drug release profiles;-Prediction of drug release;-Assessment of the fraction of drug released in time points, in which experimental data were not obtained (this is, for example, useful for estimating times at which 25%, 50%, 75%, 90%, etc. of the drug is released from a DDS);-As a quantitative drug profile analysis tool in designing and evaluating DDSs and formulations or studying drug release kinetics;-Comparison of drug release profiles;-Model-dependent analysis of drug release mechanism from DDSs (for example, when the Korsmeyer–Peppas or the Peppas–Sahlin model is applied to the first 60% of the drug release profile and the interpretation of the drug release mechanism is interpreted via the fitted model parameters).

For any application, the condition is that the model fits the raw dissolution data well, although the exact criteria for determining this are not well established. For the first four applications in the above list, any model can be used. For comparison of drug release profiles, the utilization of model-dependent methods of drug release profile comparison is not the first choice. Usually, the model-independent f_2_ similarity factor is used. However, when the f_2_ similarity factor cannot be used, a mathematical model can be fitted to drug dissolution data and the estimated model parameters can then be used for the comparison of drug dissolution profiles in combination with appropriate statistical approaches. Generally, any mathematical model can be used for this purpose; however, the FDA guidelines recommend using a model with no more than three model parameters [13,14]. On the other hand, the EMA guidelines are not specific in this manner [14]. For the assessment of the drug release mechanism, most commonly the Korsmeyer–Peppas and/or the Peppas–Sahlin models are applied to the first 60% of the drug release data, as the theory behind these models is considered to be applicable only for this dissolution data range, according to the literature [1,2,3,4,6,13,14,15,16,17,18,19,20,21,22,23,24,25,26,27,28,29,30,31,32,33,34,35,36,37,38]. In addition, the Weibull model can also be utilized in model-dependent analyses of the drug release mechanisms [12].

Several drug release profiles obtained from HPMC-based matrix tablets have been successfully analysed via mathematical models in previous studies, some of the more recent ones are mentioned in the references [31,33,35,37,39,40,41]. Generally, the Higuchi model, the Korsmeyer–Peppas model, and the Peppas–Sahlin model seem to be most appropriate for/applicable to drug release from HPMC-based matrix tablets [42,43]. However, other models, like the Zero–order model, the First–order model, the Hixson–Crowell model, the Hopfenberg model, the Baker–Lonsdale model, and the Weibull model, have also been studied in this respect [31,33,35]. A comprehensive and diverse set of mathematical models has not yet been applied to a broad range of different controlled drug release profiles based on an HPMC matrix system. The existing literature does not focus on how well mathematical models visually match dissolution data, even though fitting models can lead to varying degrees of accuracy in different parts of the drug release profile. Differences in how accurately the fitted models match the experimental dissolution data in specific sections of the drug release profile can result in inaccurate estimates of burst release or lag time at the beginning of the drug release profile, as well as inaccurate predictions of drug release towards the end of the modelled profile. In published studies thus far, mainly single point estimates of goodness of fit criteria for model fit were used, such as the coefficient of determination (Rsqr, R^2^, or COD), the adjusted coefficient of determination (Rsqr_adj or $R_{adjusted}^{2}$), or the Akaike Information Criterion (AIC). These only provide a general overview of the model fit and do not provide any information about a mathematical model’s performance in different sections of the drug release profile. Therefore, the reported goodness of fit criteria results should perhaps be complemented by a visual assessment of model fit, which should also be suitably commented on. Interestingly, there is little attention given to the residual sum of squares (RSS) as a goodness of fit criterion, although this is, in fact, the statistical parameter which is being minimized during the fitting of model parameters to raw dissolution data [22,25,28,30,38,44]. In addition, there is little to no consideration put in previous studies to how well individual mathematical models are able to adapt to varying drug release profiles observed within the same batch due to intertablet drug release variation. Simply fitting mathematical models to average drug release profiles does not give sufficient insight into the performance of mathematical models regarding their ability to model different drug release profiles stemming from intertablet drug release variability.

The goal of the present study is to critically evaluate the performance of a comprehensive and diverse set of mathematical models available in DDSolver [44] in modelling drug release from HPMC-based matrix systems via fitting the models to a comprehensive collection of different carvedilol controlled-release profiles generated in a previous study [45]. Different carvedilol release profiles were generated by analysing directly compressed HPMC-based hydrophilic matrix tablets using different selected water-soluble and water-insoluble fillers/modulators of carvedilol release (simply referred to herein as ‘fillers’). A comprehensive number of data sampling points for each formulation was used to experimentally describe the carvedilol release profiles with high accuracy. The generated carvedilol release profiles were already analysed using Local (Weighted) Regression (LOESS or LOWESS) [46,47] as a general fitting technique, where an estimation of lag time and/or burst release was conducted where applicable, among other things [45]. This LOESS analysis performed in the previous study aided in the comparative evaluation of the performance of different applied mathematical models from the viewpoint of estimating burst release and lag time. The present study aims to address some of the shortcomings of the previously published studies by complementing the overall point estimate goodness of fit results with visual analysis of model fit, considering the model fit in different sections of the drug release profile, and assessing the models’ ‘fitting flexibility’ in the context of intrabatch drug release variability. The RSS was used as a single point estimate of model fit, i.e., goodness of fit criterion, because its performance was more in line with the visual assessment of model fit than the R^2^ or the $R_{adjusted}^{2}$, which were considered initially. The model fitting was not performed just on the average carvedilol release profiles but independently on each of the four carvedilol release profiles generated per formulation. Prior to model fitting, the approximate end-point of carvedilol release was objectively determined using a paired *t*-test to identify the relevant experimental dissolution data to be used in model fitting for formulations from which practically all carvedilol was released in less than 24 h (time of dissolution analysis). In addition, model fitting was also independently performed for experimental dissolution data up to app. 60% of carvedilol released, because some formulations exhibited such extensive intrabatch carvedilol release variability that even using a paired *t*-test for dissolution end-point estimation did not guarantee avoidance of a plateau of carvedilol release for all individually tested tablets. This plateau of carvedilol release hindered model fitting for some models. In both sets of model fitting, that is, in fitting of models to the entire relevant carvedilol release profiles as determined by the paired *t*-test, and in fitting of models to carvedilol release data up to app. 60% of carvedilol released, special attention was put into the initial section of carvedilol release up to t = 60 min, which proved important for the determination of burst release or lag time. Finally, established mathematical models for analysing the mechanism of drug release from hydrophilic matrix systems were applied to generated carvedilol release profiles to elucidate the mechanism of carvedilol release from the studied formulations. Only carvedilol release data up to app. 60% of carvedilol released were used in the model-dependent analysis of the carvedilol release mechanism, as only this data range should be considered according to the literature [1,2,15,16,48,49,50].

The extensive scope of our study, encompassing numerous mathematical models applied to a diverse range of HPMC-based controlled-release drug profiles, represents a significant contribution to previously published papers of a similar topic. By integrating a general goodness of fit criterion with meticulous visual examination and inspection of fitted model parameters related to burst release and lag time, we have enhanced the accuracy of model selection. Additionally, our investigation into the models’ fitting flexibility in the context of intrabatch drug release variability adds further depth to the existing body of literature on this subject.

## 2. Materials and Methods

### 2.1. Materials

Tablets from which carvedilol release profiles were generated were prepared and analysed in a previous study, where an in-depth description of tablet manufacturing is given [45]. Carvedilol (10 *w*/*w*% of tablet weight) was used as a drug substance and METHOCELᵀᴹ K15M Premium (HPMC 2208 with nominal viscosity of 17,700 mPa∙s; 15 *w*/*w*% of tablet weight) was used as a hydrophilic matrix-forming agent. Different carvedilol release profiles were generated by varying different fillers in the formulation constituting 73.4 *w*/*w*% of tablet weight. Colloidal silicon dioxide (AEROSIL^®^ 200 Pharma; 0.3 *w*/*w*% of tablet weight) and magnesium stearate (1.3 *w*/*w*% of tablet weight) were used as glidant and lubricant, respectively. Formulations differed among each other only by the type or grade of the filler used in individual formulations and can therefore be identified by the filler used in the formulation, as follows:
Water-soluble fillers
○Two grades of polyethylene glycol/PEG (Polyglykol^®^ 4000 P, Polyglykol^®^ 8000 P);○Polyethylene oxide/PEO (Polyoxᵀᴹ WSR N-80);○Two grades of povidone (Kollidon^®^ 25, Kollidon^®^ 90 F);○Four grades of mannitol (C*Pharm Mannidex 16700, Pearlitol^®^ 160C, Parteck^®^ M 100, Parteck^®^ M 200);○Five grades of lactose monohydrate (Lactochem^®^ Crystals, Lactochem^®^ Fine Powder, SuperTab^®^ 11SD, FlowLac^®^ 100, Tablettose^®^ 70);○Sucrose (Granulated sugar N°1 600);○Maltodextrin (Glucidex^®^ 19).Water-insoluble fillers
○Two grades of anhydrous dibasic calcium phosphate/DCP (Di-Cafos^®^ A12, Emcompress^®^ Anhydrous);○Two grades of microcrystalline cellulose/MCC (Avicel^®^ PH-102, Avicel^®^ PH-200);○Ethylcellulose/EC (Ethocelᵀᴹ Standard 20 Premium);○Two samples of pregelatinized starch of the same grade but different particle sizes (Starch 1500^®^ sample with smaller particle size (↓PS), Starch 1500^®^ sample with larger particle size (↑PS)).

### 2.2. Methods

#### 2.2.1. Compression Mixtures and Tablet Preparation

The compression mixtures were prepared in a previous study on a laboratory scale using a biconical mixer. The 12 mm diameter round tablets were directly compressed using a Killian Pressima rotary tablet press.

#### 2.2.2. Carvedilol Release Profiles

The carvedilol release profiles were generated in a previous study. Dissolution analysis was performed using a method previously described by Košir et al. [41]. A dissolution apparatus type 2 with paddles, flow-through cuvettes and autosampler, 900 mL of acetate buffer solution (pH = 4.5) per vessel, dissolution media temperature of 37 °C ± 0.5 °C and sinkers to keep tablets at the bottom of the vessel were used. The amount of carvedilol released was spectrophotometrically determined at 285 nm from the measured absorbance and calculated using a calibration curve prepared in advance. Four tablets per formulation/experiment were analysed. The preselected time points for sampling were the same for all experiments and were as follows: start—0.5 h every 10 min, at 45 min, 1 h–6 h every 30 min, 6 h–24 h every 60 min.

The dissolution data used in fitting mathematical models are presented in Appendix A. Visualisations of obtained carvedilol release profiles are available in the original paper of the previous study [45].

#### 2.2.3. Determination of the Approximate End-Point of Carvedilol Release Using a Paired *t*-Test

A simple paired *t*-test was used to statistically determine a significant difference in the average fraction of carvedilol released for each of two consecutive time/data points to determine the relevant experimental carvedilol dissolution data range to consider for model fitting. The calculation was performed using Microsoft Excel. A *p*-value greater than 0.05 together with individual carvedilol release greater than 90% from each of the four tested tablets was used as a default criterion for a majority of formulations; the first time/data point, which demonstrated a paired *t*-test *p*-value greater than 0.05 in comparison to the previous time/data point, and at the same time demonstrated greater than 90% of carvedilol released from all tested tablets, was considered as the last time/data point to be used for fitting of mathematical models to dissolution data. In the case of mannitol formulations (the C*Pharm Mannidex 16700 formulation, the Pearlitol^®^ 160C formulation, the Parteck^®^ M 100 formulation and the Parteck^®^ M 200 formulation), a *p*-value greater than 0.15 was used and in case of povidone formulations (the Kollidon^®^ 25 formulation and the Kollidon^®^ 90 F formulation), a *p*–value greater than 0.20 was used as a criterion for determining the last time/data point to consider for model fitting due to larger intertablet carvedilol release variability observed in these formulations.

#### 2.2.4. Fitting of Mathematical Models to Carvedilol Release Data Using DDSolver and Overall Comparison of Model Fit

DDsolver was used for fitting mathematical models to experimental dissolution data, as it is a useful and free tool for drug dissolution data analysis, which works as a plugin for Microsoft Excel. Among other functionalities, it can be used to fit different mathematical models to dissolution data using a nonlinear least-squares curve-fitting technique, minimizing the sum of squares (SS) difference between the measured dissolution data points and model-predicted data points. It contains all the most frequently used mathematical models and some variations of models with model parameters for estimating lag time (the delay of the onset of drug release) or burst release (the uncontrolled and rapid dissolution and release of the drug from the DDS’s surface at the beginning of the drug release profile). The mathematical models available in DDSolver and utilized in the present study are presented in Table 1. The fitting of mathematical models to experimental dissolution data was performed individually for each of the four tested tablets per formulation. A summary of model fitting for each tested formulation is available in attachments, i.e., Appendix A of this article [10,25,30,31,44].

For an overall comparison of model fit, the residual sum of squares (RSS) was used as a goodness of fit criterion, as this is the statistical parameter which is minimized during model fitting in DDSolver. To assess how well the mathematical models match the early dissolution data, crucial for capturing burst release or lag time, the RSS was independently calculated for dissolution data points up to and including *t* = 60 min. This calculation was done independently for models fitted to the entire relevant dissolution data of each formulation and models fitted specifically to the dissolution data up to app. 60% of carvedilol released. All the RSS were calculated in MS Excel from observed and predicted dissolution results using equation 1, where *n* is the number of experimental dissolution data points used in model fitting, *y_i_*_,*observed*_ are the experimentally measured fractions of carvedilol released, and *y_i_*_,*predicted*_ are the model-predicted fractions of carvedilol released.
(1)$$RSS=\sum_{i=1}^{n}{\left(y_{i,\;observed}-y_{i,\;predicted}\right)}^{2}$$

The RSS results for fitting of models to the entire relevant carvedilol dissolution data, that is, up to the approximate end-point of carvedilol release as determined by the paired *t*-test, are presented in Appendix A (RSS results for the entire utilized dissolution data range in model fitting) and Appendix A (RSS results up to *t* = 60 min of the utilized dissolution data range in model fitting). The RSS results for fitting of models to carvedilol release profiles up to app. 60% of carvedilol released are presented in Appendix A (RSS results for the entire utilized dissolution data range in model fitting), and Appendix A (RSS results up to *t* = 60 min of the utilized dissolution data range in model fitting). In all cases, results are presented as average value ± one standard deviation.

In addition to the RSS results, the Pearson correlation coefficient (R), the coefficient of determination (R^2^), and the adjusted coefficient of determination ($R_{adjusted}^{2}$) are reported in the model fitting summaries available in attachments, i.e., Appendix A of this article. All of these (R, R^2^, and $R_{adjusted}^{2}$) were manually calculated in MS Excel from DDSolver’s output of model-predicted fractions of carvedilol release and original experimental dissolution data points. The Rs were calculated using MS Excel’s PEARSON function and the $R_{S}^{2}$ were calculated using the RSQ function. The $R_{adjusted}^{2}$ results were calculated using Equation (2), where *n* is the number of experimental dissolution data points used in model fitting and *p* is the number of model parameters.
(2)$$R_{adjusted}^{2}=1-\frac{n-1}{n-p}\cdot \left(1-R^{2}\right)$$

#### 2.2.5. Selection of Best-Performing Models

To choose the best-performing mathematical models from the pool of models fitted to dissolution data, several criteria were employed. The criteria used were as follows:
(1)The RSS rankingsThe RSS was used to initially rank the models, from the one which fitted the experimental dissolution data best, to the worst performing one within each formulation.(2)Criteria for visual assessment of model fitSeveral criteria were used for the visual assessment of model fit and the selection of better-suited models from less-suited ones was based on the interpretation of the fitted model parameters describing lag time or burst release in relation to experimental observations. These criteria were used on top of the initial RSS rankings to select the best-performing mathematical models from the pool of tested models. The mentioned criteria used were as follows (also see ‘Criteria for visual assessment of model fit’ in attachments, i.e., Appendix A for examples):
Progression of drug releaseA mathematical model which predicts a maximum of drug released before the last experimentally tested time point in the data set and afterwards demonstrates a significantly lower fraction of drug released in further successive time points is inferior to a model with a similar RSS result which predicts a progressively higher fraction of drug released throughout the drug release profile from *t* = 0 to *t* = max in the studied dissolution data range.The ability of the mathematical model to reproduce the sigmoid shape of a drug release profile from the experimental dissolution dataIf the experimental dissolution data clearly demonstrate the sigmoid shape of a dissolution profile, a mathematical model which is capable of reproducing/following this sigmoid shape is superior to a model with a similar RSS result which is not able to demonstrate a sigmoid shape.Matching indications of burst release or lag time (mathematical model vs. experimental dissolution data)If the experimental dissolution data indicate a possible burst release or lag time, this should be matched by the fitted mathematical model’s indication of the same two phenomena. The model which fails to do so is considered inferior to the model which matches the experimental dissolution data up to *t* = 60 min more accurately, where both phenomena can be observed.Uniformity of model fitIf two mathematical models demonstrate a similar overall fit to experimental dissolution data, i.e., RSS result, and the first one demonstrates a more uniform fit throughout the entire drug release profile than the second one (for example, the second model demonstrates a similar fit to the first one throughout the majority of the drug release profile except at the beginning, etc.), the model which demonstrates a more uniform fit throughout the entire drug release profile is considered superior.(3)Choosing representative models from model groupsWhenever possible, a single mathematical model was selected as a representative model from a model group, unless there was no clear indication of which model in a certain model group was better suited than the other one(s) based on the interpretation of fitted model parameters describing lag time or burst release in relation to experimental dissolution results or based on the visual examination of model fit. Therefore, if the experimental results showed no clear indication of burst release or lag time, a model with a negligible low absolute value of *T_lag_* or *F*_0_ was not penalized in relation to a model without a *T_lag_* or F_0_ term in it, and could still be selected as a candidate for predicting/representing carvedilol release. In the Peppas–Sahlin model group, the Peppas–Sahlin_1 and the Peppas–Sahlin_1 with *T_lag_* models with an *m* value of 0.45 or lower were preferred over the Peppas–Sahlin_2 and the Peppas–Sahlin_2 with T_lag_ models, as the latter case is a less appropriate option for the produced tablets according to their aspect ratio, that is, the ratio between tablet diameter and tablet thickness (see Section 2.2.6 for explanation).

In order to clearly mark the most suited mathematical models, which were chosen from the pool of tested models according to the above criteria, the RSS model fit results of fitting the models to the entire relevant dissolution data range, as determined by the paired *t*-test, and the RSS model fit results of fitting the models to the dissolution data range up to app. 60% of carvedilol released were colour-coded as follows:-Green colour, bolded: the best-performing mathematical model(s) chosen according to the above criteria, i.e., the primary-choice or the first-choice model(s).-Light green colour, bolded: the near best-performing mathematical model, i.e., the near primary-choice or near first-choice model, whose performance is slightly worse but very close to the best-performing one, and is at the same time significantly better than the secondary-choice/second-choice model’s performance.-Light gold colour, bolded: secondary-choice/second-choice model, whose performance is significantly worse than the best-performing one, but is still clearly good enough to make the model practically usable for predicting carvedilol release with significant accuracy in comparison to other tested models.-Orange colour, bolded: tertiary-choice/third-choice model, whose performance is significantly worse than the secondary-choice/second-choice one, but still clearly good enough to make the model practically usable for predicting carvedilol release with reasonable accuracy in comparison to other tested models.-Grey colour, bolded: mathematical model, whose RSS result was similar to the best-performing model or near best-performing one, secondary-choice/second-choice or tertiary-choice/third-choice one, but its fitted model parameters indicate lag time or burst release are not in line with experimental dissolution results, i.e., they significantly differ from experimental observations of carvedilol release or the model is not chosen as a representative model in the model group.-No colour-coding: mathematical models whose model fit was significantly worse and/or less suited for carvedilol release profiles representation than the above ranked models according to the utilized criteria.

A summary of model performance for fitting of models to the entire relevant carvedilol release data is presented in Appendix A (individual models’ performances) and Figure 1 (model groups’ performances). A summary of model performance for fitting of models to carvedilol release data up to app. 60% of carvedilol released is presented in Appendix A (individual models’ performances) and Figure 3 (model groups’ performances). More detailed RSS results of models, together with the above-explained colour-coding of results, are presented in Appendix A (fitting of models to the entire relevant carvedilol release data) and Appendix A (fitting of models to carvedilol release data up to app. 60% of carvedilol released).

In the case of the RSS model fit results applied to time/data points up to and including *t* = 60 min, only the RSS value was used to determine the best-performing, the near best-performing, the second-choice, and the third-choice models without any other criteria. The colour-coding used was the same as stated above. A summary of model performance up to and including *t* = 60 min is presented in Appendix A (individual models’ performances) and Figure 2 (model groups’ performances) for fitting of models to the entire relevant carvedilol release data, and Appendix A (individual models’ performances) and Figure 4 (model groups’ performances) for fitting of models to carvedilol release data up to app. 60% of carvedilol released. More detailed RSS results applied to time/data points up to and including *t* = 60 min together with colour-coding are presented in Appendix A (fitting of models to the entire relevant carvedilol release data) and Appendix A (fitting of models to carvedilol release data up to app. 60% of carvedilol released).

**Figure 2 pharmaceutics-16-00498-f002:**
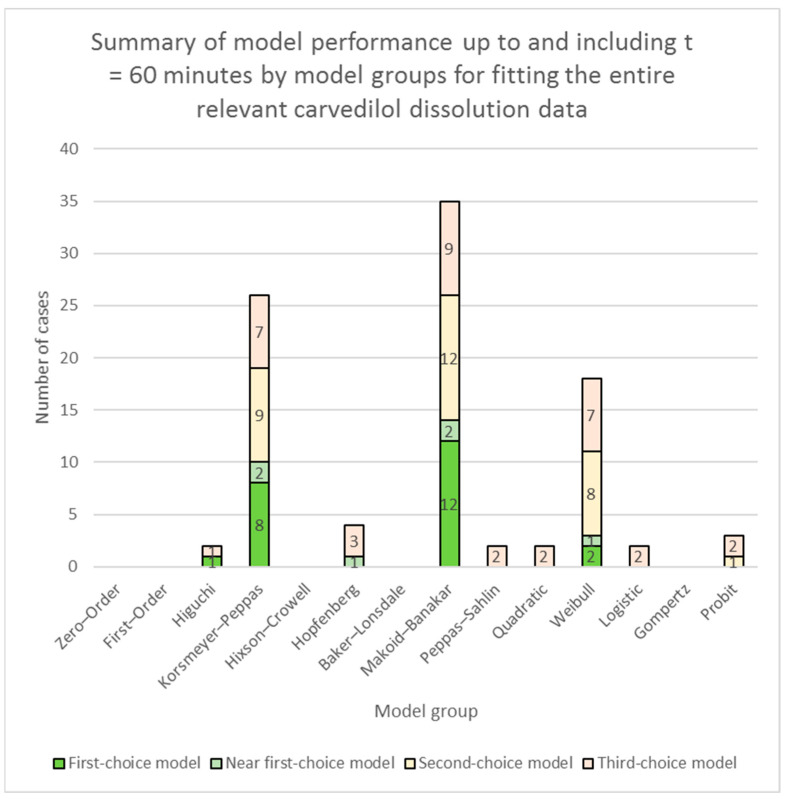
Summary of model performance up to and including *t* = 60 min by model groups (see Table 1) for fitting the entire relevant carvedilol dissolution data. The figure depicts the number of times one or more models in a model group performed as a first-choice model, a near first-choice model, a second-choice model, or a third-choice model. The height of each column depicts the total number of times one or more models from each model group were chosen as candidates for carvedilol release modelling.

**Figure 3 pharmaceutics-16-00498-f003:**
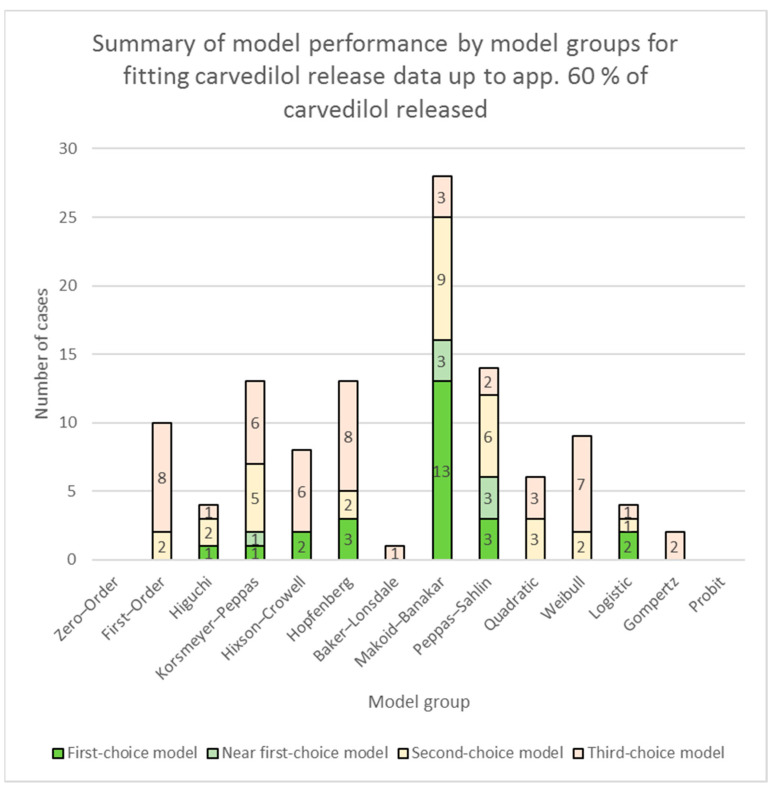
Summary of model performance by model groups (see Table 1) for fitting carvedilol release data up to app. 60% of carvedilol released (for the Polyglykol^®^ 4000 P and the Polyglykol^®^ 8000 P formulations, dissolution data up to app. 75% of carvedilol released were used in order to include enough dissolution analysis data points to fit all the available models in DDSolver; for the Parteck^®^ M 100 formulation, dissolution data up to app. 70% of carvedilol released were used due to the same reason). The figure depicts the number of times one or more models in a model group performed as a first-choice model, a near first-choice model, a second-choice model, or a third-choice model. The height of each column depicts the total number of times one or more models from each model group were chosen as candidates for carvedilol release modelling.

**Figure 4 pharmaceutics-16-00498-f004:**
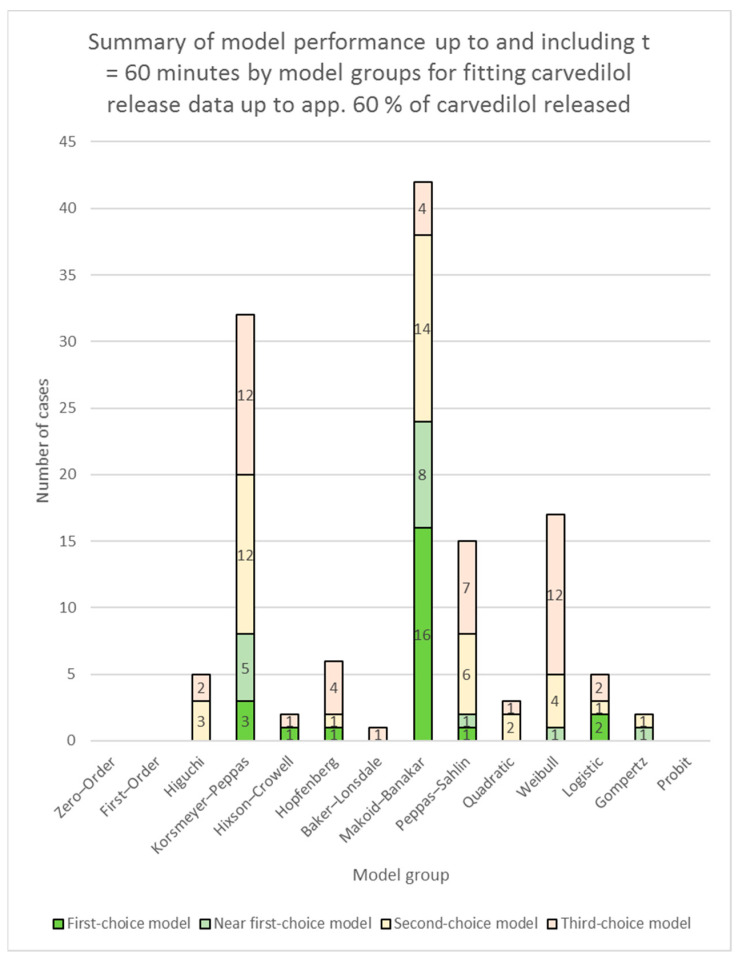
Summary of model performance up to and including *t* = 60 min by model groups (see Table 1) for fitting carvedilol release data up to app. 60% of carvedilol released (for the Polyglykol^®^ 4000 P and the Polyglykol^®^ 8000 P formulations, dissolution data up to app. 75% of carvedilol released were used in order to include enough dissolution analysis data points to fit all the available models in DDSolver; for the Parteck^®^ M 100 formulation, dissolution data up to app. 70% of carvedilol released were used due to the same reason). The figure depicts the number of times one or more models in a model group performed as a first-choice model, a near first-choice model, a second-choice model, or a third-choice model. The height of each column depicts the total number of times one or more models from each model group were chosen as candidates for carvedilol release modelling.

#### 2.2.6. Analysis of Carvedilol Release Mechanism Using the Korsmeyer–Peppas and the Peppas–Sahlin Models

To analyse the mechanism of carvedilol release, the following models were used: the Higuchi model, the Higuchi with *T_lag_* model, the Higuchi with F_0_ model, the Korsmeyer–Peppas model, the Korsmeyer–Peppas with *T_lag_* model, the Korsmeyer–Peppas with *F*_0_ model, the Peppas–Sahlin_1 model, and the Peppas–Sahlin_1 with *T_lag_* model. Carvedilol dissolution data for up to app. 60% of carvedilol released were used, except for the Polyglykol^®^ 4000 P, the Polyglykol^®^ 8000 P, and the Parteck^®^ M 100 formulations, where in order to fit the Peppas–Sahlin_1 with *T_lag_* model, an additional experimental dissolution data point had to be considered to facilitate the fitting of the model, resulting in up to app. 75% of carvedilol released dissolution data being used for the Polyglykol^®^ 4000 P and the Polyglykol^®^ 8000 P formulations, and up to app. 70% of carvedilol released dissolution data being used for the Parteck^®^ M 100 formulation. If any of the Higuchi models demonstrated a strong fit with the experimental dissolution data—selected as the best-performing, near best-performing, second-choice, or third-choice models for carvedilol dissolution data up to app. 60% of carvedilol released—this suggested that carvedilol release followed the Fickian diffusion mechanism. For the Korsmeyer–Peppas models, the carvedilol release mechanism was analysed according to the value of the diffusional exponent *n* as defined for a cylindrical shape, considering that the shape of the produced tablets was cylindrical (see Table 2). For the Peppas–Sahlin_1 models, the *m* value of 0.45 or lower was used (fitted by DDSolver), as this was an appropriate *m* value according to the aspect ratio of produced tablets (2a/l, where 2a is the tablet diameter and l is the tablet thickness), which ranged from app. 1.9 to app. 3.1, considering the tablet diameter of 12 mm and tablet average thickness data ranging from 3.86 to 6.25 mm. The constants *k*_1_ (the constant related to the Fickian kinetics) and *k*_2_ (the constant related to Case–II relaxation kinetics) were interpreted as relative contributions of the Fickian diffusion and the Case–II relaxation mechanisms to the overall carvedilol release, respectively. Models with lower RSS results and better visual fit to dissolution data were considered more important for the model-dependent interpretation of carvedilol release mechanisms than models with higher RSS values [1,2,15,16,48,49,50,51,52].

In addition to the Higuchi, Korsmeyer–Peppas, and Peppas–Sahlin models, the Weibull model can also be employed for a model-dependent interpretation of drug release mechanisms [12]. However, this model was not considered in this article, as we opted for a more traditional and commonly reported approach to the model-dependent interpretation of drug release mechanisms.

A summary of model fit (RSS results with colour-coding) and fitted model parameters utilized in the analysis of the mechanism of carvedilol release from studied formulations is presented in Table 3 and Table 4.

## 3. Results and Discussion

### 3.1. Fitting of Models to the Entire Relevant Carvedilol Dissolution Data

Considering the models’ performance in fitting the entire relevant carvedilol release data as determined by the paired *t*-test, the Makoid–Banakar model proved to be the most useful, followed closely by the Hopfenberg with T_lag_ model (see Appendix A). Other models worth mentioning are the Peppas–Sahlin_1 model, the Korsmeyer–Peppas with *F*_0_ model, the Weibull_3 model, the Weibull_2 model, the Hixson–Crowell with *T_lag_* model, and the Quadratic model. It seems that with these models, carvedilol release can be satisfactorily modelled for the vast majority of tested formulations. Out of these models, the Hixson–Crowell with *T_lag_* model can be omitted as it can be replaced by the Hopfenberg with *T_lag_* model in all tested cases, producing an equally good or even better result. The models in the Hixson–Crowell model group are encompassed within the models in the Hopfenberg model group as evident from the models’ equations presented in Table 1. 

Looking more broadly and considering the model groups, the Makoid–Banakar, the Hopfenberg, the Weibull, the Korsmeyer–Peppas, and the Peppas–Sahlin model groups stand out as the most important ones (see Figure 1). Interestingly, models in the Korsmeyer–Peppas model group can in some cases model the entire relevant carvedilol release profile with significant accuracy, although they were developed for modelling drug release profiles only up to app. 60% of the drug released [1,2,15,16,48,49]. The Makoid–Banakar model is similar to the Korsmeyer–Peppas model, except that it contains an additional exponential term in its equation (see Table 1), which makes it more flexible than the Korsmeyer–Peppas model. It is even capable of fitting to a sigmoid-shaped drug release profile, as long there is no plateau of drug release present (see the Makoid–Banakar model’s fit for the Polyoxᵀᴹ WSR N-80 formulation in the model fitting summary; see also an example of a failure of the Makoid–Banakar model’s fit at a plateau of carvedilol release in the model fitting summary of the Kollidon^®^ 90 F formulation), which the Korsmeyer–Peppas model cannot do. Other models, such as the ones in the Weibull, the Logistic, the Gompertz, and the Probit model groups, are also capable of fitting to the sigmoid shape of a drug’s release profile and are also very flexible, but they generally did not perform nearly as well as the Makoid–Banakar model for tested formulations overall, although there were cases where some of them outperformed the Makoid–Banakar model (see Appendix A). On the other hand, the additional exponential term in the Makoid–Banakar model’s equation makes this model incapable of estimating the drug release mechanism as it can be done by analysing the fitted *n*-value, i.e., diffusional exponent of the Korsmeyer–Peppas model (see Table 1 and Table 2). 

In DDSolver, there is no variant of the Makoid–Banakar model which would contain a burst release parameter (F_0_), and in the model variant with a lag time parameter (*T_lag_*), the Makoid–Banakar with *T_lag_*, DDSolver does not permit estimating a negative *T_lag_*, which would indicate a positive *F*_0_, i.e., burst release. Hence, both models in the Makoid–Banakar model group are expected to encounter challenges when fitting drug release data in formulations that exhibit significant burst release. In previous research utilizing LOESS modelling [45], a significant positive F_0_, i.e., burst release was estimated for several formulations. The largest F_0_ was observed in batches utilizing Lactochem^®^ Crystals, Tablettose^®^ 70, and SuperTab^®^ 11SD as fillers (see % of carvedilol released at *t* = 10 min in Appendix A, and LOESS *F*_0_ estimates in Appendix A). For these three formulations, the Higuchi with F_0_ model overall performed better than the Makoid–Banakar model, and the Korsmeyer–Peppas with *F*_0_ model was also a viable choice (see Appendix A), outperforming the Makoid–Banakar model in the initial stage of carvedilol release for two out of the three mentioned formulations (see Appendix A). For longer carvedilol release profiles, which exhibit a moderate but still significant burst release (see the LOESS F_0_ estimate in Appendix A for the Emcompress^®^ Anhydrous formulation and both Starch 1500^®^ formulations), the Korsmeyer–Peppas with F_0_ model was suitable in only one out of three cases, whereas the Hopfenberg with *T_lag_* model with a negative *T_lag_* estimate, indicating burst release, was suitable in all three cases (see Appendix A and the model fitting summaries for the three mentioned formulations). In fact, the Hopfenberg with T_lag_ model was among the top performers for all formulations which released carvedilol very slowly, namely the sucrose and the maltodextrin formulations along with all the formulations using selected water-insoluble fillers (see Appendix A). Moreover, the Hopfenberg with T_lag_ model can effectively simulate a drug release plateau without contradicting the ‘progression of drug release’ guideline (see Section 2.2.5). This feature renders it quite adaptable for modelling drug release in formulations where noticeable intertablet variation is observed. This is evident from its performance in the Kollidon^®^ 90 F formulation and the mannitol formulations with the exception of the Parteck^®^ M 200 formulation (see Appendix A and the model fitting summaries for the mentioned formulations). The intertablet carvedilol release variability is significant in these formulations and any model used in these cases has to be able to model fairly different carvedilol release profiles obtained from individual tablets. In even more severe cases of intertablet drug release variability, the flexibility of the Hopfenberg model group is not enough, and models in the even more flexible Weibull model group perform better (see Appendix A and the model fitting summaries for both PVP and the mannitol Parteck^®^ M 100 and 200 formulations). 

In the most sigmoid-shaped carvedilol release profile, obtained with the Polyoxᵀᴹ WSR N-80 formulation, the Weibull_3 and the Weibull_4 models outperform the Makoid–Banakar model, although the latter is also capable of fitting to a sigmoid-shaped drug release profile (see Appendix A and the model fitting summary for the Polyoxᵀᴹ WSR N-80 formulation). Although models in the Logistic, Gompertz, and Probit model groups can also model a sigmoid profile shape, they were generally outperformed by the models in the Weibull model group (see Appendix A). 

The Quadratic model was useful in only three cases, all of which exhibited a fairly slow carvedilol release (see Appendix A). It has its place in model fitting but it proved to be prone to violating the ‘progression of drug release’ rule (see Section 2.2.5) when carvedilol release approached a plateau for any of the tested tablets. 

The models in the Peppas–Sahlin model group do not contain a term for burst release (*F*_0_), and a negative *T_lag_* fitting, indicating burst release, is not allowed in DDSolver as, for example, in the Hopfenberg with *T_lag_* model. Hence, this model group was not able to perform well in cases where significant burst release was present, at least in representing the initial section of carvedilol release. Analogously to the Korsmeyer–Peppas model, the Peppas–Sahlin model also proved useful in fitting entire relevant carvedilol release profiles in individual cases, although it was developed for modelling only up to app. 60% of the drug released [1,2,15,16,50]. It proved useful in the two lactose formulations exhibiting a moderate burst release (the Lactochem^®^ Fine Powder formulation and the FlowLac^®^ 100 formulation; see Appendix A for LOESS burst release estimations) and most of the formulations using selected water-insoluble fillers (see Appendix A).

All in all, results show that none of the models are flexible enough to fully cope with the very large intertablet carvedilol release variability observed in some cases, as is evident from the high and very variable RSS values in these cases (see Appendix A for the PVP and mannitol formulations and model fitting summaries for these formulations).

While evaluating the overall performance of models is important, it does not necessarily imply that the best overall-performing models excel in every section of drug release profiles. Examination of models’ performances up to *t* = 60 min release time, which proved to be a section of carvedilol release profiles important for describing burst release, lag time, or sigmoid onset of carvedilol release, results in a much different picture than the models’ overall performance (see Figure 2 in comparison to Figure 1, and Appendix A in comparison to Appendix A). Most notably, the Hopfenberg with *T_lag_* model and the Hopfenberg model group in general, performing very well overall, were severely outperformed by the Makoid–Banakar and the Korsmeyer–Peppas model groups in the initial section of carvedilol release. Results clearly demonstrate that in most cases the Hopfenberg model group does not even come close to the performance of the Makoid–Banakar and the Korsmeyer–Peppas model groups, except in some rare cases (see Appendix A). Results also indicate that for estimating *F*_0_ or *T_lag_*, perhaps different models should be considered than the ones excelling in the overall model performance, at least in some cases. A good example of this is the performance of the Hopfenberg with *T_lag_* model in fitting to carvedilol release data of the sucrose formulation, the maltodextrin formulation, and all of the formulations using selected water-insoluble fillers. The overall performance of the model in these cases was good (see Appendix A, and the model fitting summaries for the mentioned formulations) but the burst release was significantly overestimated in comparison to raw dissolution data (see % of carvedilol released at *t* = 10 min in Appendix A) and LOESS burst release, i.e., *F*_0_ estimates (see Appendix A). Therefore, the Hopfenberg with *T_lag_* model was in these cases also penalized during the visual ‘uniformity of model fit’ examination (see Section 2.2.5), which influenced its ranking as a first-, second-, or third-choice model in modelling the entire relevant carvedilol release profile.

### 3.2. Fitting of Models to Carvedilol Release Data up to App. 60% of Carvedilol Released

Using a paired *t*-test to determine the relevant dissolution data range for model fitting (see Section 2.2.3 for an explanation) was not enough to avoid the occurrence of a plateau of carvedilol release in the case of some individual tested tablets. This issue, hindering the fitting of some models, was present in formulations using selected water-soluble fillers, especially in the PVP and the mannitol group of formulations, where significant intertablet carvedilol release variability was observed. Using a dissolution data range up to app. 60% of carvedilol released mitigated the mentioned issue in all cases, even for formulations exhibiting severe intertablet carvedilol release variability. Three formulations, namely both PEG formulations and the Parteck^®^ M 100 formulation, released carvedilol so fast that an app. average of 60% of carvedilol released was achieved in just 45 min, yielding only four dissolution data points to be considered for model fitting. Four dissolution data points were not enough to fit all the available models in DDSolver to the experimental dissolution data. For this reason, an additional dissolution data point at *t* = 60 min was included in the mentioned formulations, yielding the necessary minimum of five dissolution data points to fit all the models. This resulted in utilizing dissolution data up to app. 75% of carvedilol released in the case of both PEG formulations and up to app. 70% of carvedilol released in the case of the Parteck^®^ M 100 formulation. None of the tested tablets reached a plateau of carvedilol release after *t* = 60 min; therefore, this action did not interfere with fitting any of the models and did not jeopardise the model performance comparison.

Fitting of models to dissolution data up to app. 60% of carvedilol released painted a somewhat different picture than fitting of models to the entire relevant carvedilol release data. The dominance of the Makoid–Banakar model and the Makoid–Banakar model group in general was further enhanced (see Appendix A, Figure 3, and Appendix A). The Makoid–Banakar model was among the top performers for all the tested formulations and a first-choice model for more than half of them. The model’s inheritance from the Korsmeyer–Peppas model and the added additional exponential term in its equation (see Table 1), which enhances its flexibility in comparison to the Korsmeyer–Peppas model, proved to be the right recipe for modelling carvedilol release profiles up to app. 60% of carvedilol released. The Makoid–Banakar model performed better with carvedilol release profiles obtained from formulations using selected water-soluble fillers as it was outperformed in only one lactose and two mannitol formulations (see Appendix A), but not by a significant margin. In the slower carvedilol release profiles obtained using sucrose, maltodextrin, or selected water-insoluble fillers, it was the best performer for the EC and Starch 1500^®^ formulations, but was outperformed by the Peppas–Sahlin model group for other formulations (see Appendix A). 

Looking at other well-performing models, the Hopfenberg with T_lag_ model was again among the top performers, like in modelling the entire relevant carvedilol release profiles. It proved suitable for both PEG formulations, three of the four mannitol formulations, sucrose and maltodextrin formulations, and all the formulations using selected water-insoluble fillers except for the Emcompress^®^ Anhydrous formulation (see Appendix A). Based on prior knowledge obtained during the previous study, there is no discernible pattern to observe in these formulations [45]. For the Polyglykol^®^ 8000 P formulation and both Parteck^®^ M formulations, the Hopfenberg with *T_lag_* model’s lag time indication is in line with previous analysis using LOESS (see the model fitting summaries for the mentioned formulations and Appendix A). Interestingly, the model did not prove to be among the top performers for the lactose formulations, mainly due to its overestimation of burst release (see Appendix A and model fitting summaries for all lactose formulations). For the longer carvedilol release profiles, obtained with sucrose, maltodextrin, and selected water-insoluble fillers, the Hopfenberg with *T_lag_* model performed generally well, although it was not the best performer in cases other than the sucrose and the maltodextrin formulations (see Appendix A). It generally indicated some burst release, in line with previous LOESS-based analysis (see the model fitting summaries for the sucrose and maltodextrin formulations and formulations using selected water-insoluble fillers, and Appendix A). Similarly to the lactose formulations, for the Emcompress^®^ Anhydrous formulation, this model was not a top performer, again due to its significant overestimation of burst release (see Appendix A and model fitting summaries for the Emcompress^®^ Anhydrous formulation). Looking at the carvedilol release profiles up to app. 60% of carvedilol released from the Hopfenberg with *T_lag_* model’s performance in general, it seems that this model has difficulties with carvedilol release profiles, where a significant curvature is present in the initial part of the release profile, followed by a fairly linear carvedilol release. The model is unable to adequately adapt to this curvature present at the beginning of some carvedilol release profiles and thus overestimates burst release. 

Models in the Hixson–Crowell model group, although worth mentioning, have no advantage over the models in the Hopfenberg model group, as the models from the first group are encompassed in the latter group (see Table 1). Therefore, it is not surprising that the Hixson–Crowell model group never outperformed the Hopfenberg model group (see Appendix A). 

The models in the Peppas–Sahlin model group were next in line according to their performance. Among the shorter carvedilol release profiles, namely the ones obtained with PEG/PEO, PVP, mannitol, or lactose as fillers, they generally did not perform well, except in some individual cases (see Appendix A). This model group performed well for the longer carvedilol release profiles obtained with sucrose, maltodextrin, and selected water-insoluble fillers, where it was the top performer for the MCC and EMCOMPRESS^®^ Anhydrous formulations (see Appendix A). It did not perform satisfactorily only with one of the two Starch 1500^®^ formulations, which exhibited significant burst release, as models in this model group do not have the ability to model burst release due to reasons explained in the previous section (see Appendix A, and the model fitting summaries for the two Starch 1500^®^ formulations). Models in this group also had no trouble fitting the initial curvature of these longer carvedilol release profiles in contrast to the Hopfenberg with *T_lag_* model. This is probably due to the fact that the models in the Peppas–Sahlin model group balance the contribution of Fickian diffusion kinetics with their *k*_1_ constant and the Case–II relaxation kinetics with their *k*_2_ constant (see Table 1) [15,16]. The first type of drug release kinetics results in a curved drug release profile and the latter in a linear one. 

The First–order model group performed well in some cases. In this group, only the First–order with *T_lag_* model proved useful (see Figure 3, Appendix A). In the group of formulations using selected water-soluble fillers, it performed well only for one of the four tested mannitol formulations, one of the five tested lactose formulations, and in the sucrose formulation, which along with the maltodextrin formulation released carvedilol considerably slower than the other formulations using selected water-soluble fillers (see Appendix A). In contrast, for the even slower carvedilol release profiles obtained using selected water-insoluble fillers, it was among the top contenders in all cases, although it was a third-choice model for all but one of them (see Appendix A). No additional rules could be discerned, except for the general utility of this model in the context of longer-duration carvedilol-releasing formulations. Like the models in the Hopfenberg and the Hixson–Crowell model groups, this model had difficulties adjusting to the curvature of carvedilol release profiles present at the beginning of carvedilol release profiles. Consequently, estimating a significant negative *T_lag_*, it overestimated burst release in all cases where it was among the chosen models. However, it performed rather well in other non-initial sections of the carvedilol release profiles (see model fitting summaries for formulations using selected water-insoluble fillers). 

The models in the Korsmeyer–Peppas model group proved useful in individual cases of formulations using selected water-soluble and selected water-insoluble fillers. These models generally did not perform well for the faster carvedilol-releasing formulations using PEG, PEO, PVP, or mannitol as fillers (see Appendix A). However, in one individual case, that is, in the Polyglykol^®^ 4000 P formulation, the Higuchi with *F*_0_ model performed as a second-choice model up to app. 75% of carvedilol released and the same model along with the Korsmeyer–Peppas with *T_lag_* model performed satisfactorily up to app. 60% of carvedilol released (see Appendix A). In all cases, a small lag time was estimated (see model fitting summaries for the mentioned formulations). These results are not that significant as only four or five dissolution data points were considered, thus leaving significant space for interpretation of different possible curvatures connecting these data points and estimating either a small lag time, no lag time or burst release, or even a small burst release before the first data point. Interestingly, for the Polyglykol^®^ 4000 P formulation in the up to app. 75% of carvedilol released dissolution data range, the Higuchi with F_0_ model performed better than the Korsmeyer–Peppas with F_0_ model, although both were identical if the value of the diffusional exponent *n* in the Korsmeyer–Peppas with F_0_ model was equal to 0.5 (see Table 1). The Korsmeyer–Peppas model group performed well for the lactose formulations (see Appendix A), which release carvedilol slower than the PEG, PEO, PVP, and mannitol formulations (see Appendix A). For these lactose formulations, the Higuchi with F_0_ model also performed well in three out of five cases (see Appendix A). Interestingly, it even surpassed the Korsmeyer–Peppas with F_0_ model’s performance in two instances (see Appendix A). For the slower carvedilol-releasing formulations using sucrose, maltodextrin, or selected water-insoluble fillers, the Korsmeyer–Peppas model group performed well, except for both STARCH 1500^®^ formulations (see Appendix A). The initial curvature of carvedilol release followed by practically linear carvedilol release present in both Starch 1500^®^ formulations proved to be too much for the limited flexibility of the Korsmeyer–Peppas model group, but not for the Makoid–Banakar model, which proved to be the best performer in both Starch 1500^®^ formulations (see Appendix A and model fitting summaries for the mentioned formulations). Lastly, it is important to note that instances where the Higuchi with *F*_0_ model outperformed the Korsmeyer–Peppas with *F*_0_ model should not be considered realistic. The Korsmeyer–Peppas with *F*_0_ model, with a diffusional exponent (*n*) value of 0.5, is identical to the Higuchi with *F*_0_ model. Therefore, this discrepancy, where the Higuchi with *F*_0_ model outperformed the Korsmeyer–Peppas with F_0_ model, likely originated from inaccuracies in the model parameter fitting process using DDSolver, rather than reflecting true differences in the models.

Next in line was the Weibull model group, which contains flexible models able to model various drug release profile shapes. It proved useful in some faster carvedilol-releasing formulations, that is, in both PEG formulations, the PEO formulation, and three out of four mannitol formulations (see Appendix A). Interestingly, it did not perform well in any of the PVP formulations (see Appendix A and model fitting summaries for both PVP formulations). In lactose formulations or in any other formulations, which release carvedilol even more slowly than the lactose formulations, it did not perform well (see Appendix A). 

The last model group worth mentioning is the Quadratic model group. There is no real mechanistic theory which would relate the Quadratic model with drug release from matrix systems. However, this model proved useful in some individual cases, where formulations released carvedilol fairly fast, like the PEG, PVP, and mannitol formulations (see Appendix A). However, there is no apparent rule here to extract from the data regarding in which cases the Quadratic model could be useful. 

Some of the other models available in DDSolver were also useful in some individual cases, but are not worth mentioning as they came in handy only in some rare occurrences and could mostly be replaced by other models mentioned thus far.

Considering the models’ performances up to *t* = 60 min, the Makoid–Banakar model group remained the top performer, followed by the Korsmeyer–Peppas model group, and then the Peppas–Sahlin and the Weibull model groups (see Figure 4, Appendix A). Other models are not really worth mentioning as they have seldom proved useful. The Makoid–Banakar model was the best performer even for the Lactochem^®^ Crystals, the SuperTab^®^ 11SD, and the Tablettose^®^ 70 formulations (see Appendix A), although these formulations exhibited significant burst release (see Appendix A). A burst release (*F*_0_) cannot be estimated by the Makoid–Banakar model, as it does not contain a term for burst release or lag time in its equation (*F*_0_ or *T_lag_*, respectively), nor is the Makoid–Banakar with *T_lag_* model permitted to estimate a negative *T_lag_* (a sign of a positive *F*_0_ and therefore burst release) in the DDSolver software. The reason why the Makoid–Banakar model was able to perform so well even in these cases lies in its already-mentioned flexibility. The only exception where the Makoid–Banakar model group did not perform on the top level is the mannitol formulation with C*Pharm Mannidex 16700 (see Appendix A), but only because the model did not fit well in the case of one of the four tested tablets of this formulation (see the model fitting summary for the C*Pharm Mannidex 16700 formulation). In this case, the even more flexible models of the Weibull, the Logistic, and the Gompertz groups outperformed the models in the Makoid–Banakar model group (see Appendix A). 

In order to estimate burst release detected in the previous study via LOESS for the three mentioned lactose formulations (the Lactochem^®^ Crystals, the SuperTab^®^ 11SD, and the Tablettose^®^ 70 formulations), one has to consider the Korsmeyer–Peppas with *F*_0_ and the Higuchi with *F*_0_ models, as they were among the top performers in modelling dissolution data up to *t* = 60 min for these formulations (see Appendix A and the model fitting summaries for the mentioned lactose formulations). Furthermore, The Korsmeyer–Peppas model group generally performed well in all the slower carvedilol-releasing formulations, that is, in the sucrose, the maltodextrin, and the selected water-insoluble fillers-based formulations (see Appendix A). The Peppas–Sahlin model group similarly performed well for the same formulations except for the two Starch 1500^®^ formulations (see Appendix A). It generally did not perform well for the formulations releasing carvedilol faster, except in the case of one lactose formulation, where it performed as a third-choice model group (see Appendix A). 

The Weibull model group proved useful in formulations releasing carvedilol faster, namely in the PEG, PEO, the slower carvedilol-releasing PVP formulation (the Kollidon^®^ 90 F formulation), and in three out of the four mannitol formulations (see Appendix A).

In conclusion, the number of parameters in each mathematical model (see Appendix A) appears to have influenced the models’ ability to fit the raw dissolution data. However, there are numerous instances where mathematical models with three or four parameters—the models with the maximal number of parameters in the utilized pool of models—did not perform well. This suggests that the mathematical structure of the models themselves played a crucial role in their ability to fit the raw dissolution data.

Some of the model parameters fitted by DDSolver lack physical grounding and cannot be utilized in the model-dependent interpretation or comparison of drug release. Examples include negative fitted values of lag time or burst release in some models (although a positive burst release value can be manually estimated from a model with a negative lag time, and vice versa), as well as negative model constants, such as those observed in the Peppas–Sahlin model group in some cases. DDSolver lacks all suitable variations of mathematical models regarding burst release or lag time for all models: a basic model with no *F*_0_ or *T_lag_* parameter, a variation of the model with *T_lag_*, and a variation of the model with *F*_0_. Additionally, the software exhibits inconsistency in applying suitable constraints for fitting lag time and burst release parameters, which should be ≥0, and model constants like *k*_1_ or *k*_2_ for the Peppas-Sahlin models (which should also be ≥0). Despite the presence of “unphysical” parameter values, the fitted models can still be useful in some cases if they fit well. For example, they can be used to calculate the fraction of drug released at time points where no experimental data were acquired, or to calculate the time at which a certain percentage of the drug has been released (e.g., T_25%_, T_50%_, T_75%_, T_90%_, etc.). 

Generally, the LOESS estimations of lag time and burst release from the previous study (Appendix A) [45] seem to be more useful in comparing different formulations considering these two phenomena, as the lag time or burst release estimate was performed using the same modelling approach. Using different available mathematical models, although they can be used for lag time or burst release estimations, one can be in doubt if it makes sense to compare formulations from a lag time and burst release viewpoint using different mathematical models for different formulations. In addition, a smaller set of experimental dissolution data points in the initial stage of drug release is probably better to consider than the whole set of data points, if one decides to use mathematical models in lag time or burst release estimations. For improved accuracy in estimating burst release or lag time using the LOESS approach, the setting that determines the number of data points considered in creating local models could be adjusted to a lower, suitable value. In this way, the desired higher accuracy of burst release or lag time estimation could be achieved without doubting the validity of the formulations’ comparison since the same approach would be consistently applied across all formulations. The only hypothetical cases where the mathematical models could be superior to LOESS are perhaps the sigmoid-shaped drug release profiles. In these cases, the LOESS methodology could result in lag time estimation where there really is none and a suitable mathematical model capable of modelling a sigmoid curve, like the Makoid–Banakar model, the Weibull model, and others could prove superior. The Weibull_1 and the Weibull_4 models available in DDSolver even have the possibility of estimating lag time in combination with a sigmoid-shaped drug release profile. However, for the lag time estimation to be valid, the model fit would have to be very good, which cannot always be achieved based on the presented examples in this study. In all cases, it helps to have as many experimental dissolution data points available as possible, at least in the initial stage of drug release. 

### 3.3. Model-Dependent Estimation of the Mechanism of Carvedilol Release

A summary of model data, which was used in the model-dependent analysis of the carvedilol release mechanism, is presented in Table 3 and Table 4.

Table 3 presents the RSS results of both Higuchi and Korsmeyer–Peppas models utilized in the model-dependent interpretation of carvedilol release mechanisms. Additionally, it includes the diffusional exponent (*n*) values of the fitted Korsmeyer–Peppas models, which were used in the interpretation of carvedilol release mechanisms in accordance with information presented in Table 2, as stated in Section 2.2.6. Higuchi models served as an indicator of possible Fickian diffusion-based release if any exhibited a strong fit compared to Korsmeyer–Peppas models, as discussed in Section 2.2.6. Models with a lower RSS result were considered as more important, since this indicated a better fit of the model. The data in Table 3, which were considered in the model-dependent interpretation of carvedilol release mechanisms, are bolded. The colour-coding of the results in Table 3 is consistent with the one presented in Section 2.2.5. 

Table 4 presents RSS results and model constants of utilized Peppas–Sahlin models in the model-dependent interpretation of the carvedilol release mechanisms. Only non-negative values of *k*_1_ and *k*_2_ constants were considered. As outlined in Section 2.2.6, the constant *k*_1_ is related to Fickian kinetics and constant *k*_2_ to Case–II relaxation kinetics. If both constants *k*_1_ and *k*_2_ were > 0 and the model’s RSS was relatively low, indicating a good fit, fitted values of *k*_1_ and *k*_2_ constants were interpreted as relative contributions of Fickian diffusion and Case–II relaxation mechanisms to overall carvedilol release, respectively. Similarly to Higuchi and Korsmeyer–Peppas models, Peppas–Sahlin models with lower RSS results and better visual fit to dissolution data were prioritized for the model-dependent interpretation of carvedilol release mechanisms. The colour-coding of results in Table 3 is consistent with the one presented in Section 2.2.5.

The fit of each model mentioned in the analysis was analysed for each individual tested tablet in terms of the RSS result of model fit and visual examination of model fit. In addition, key model parameters used in the interpretation of the carvedilol release mechanism were also considered for each individual tested tablet. Negative estimations of k_1_ or k_2_ for the Peppas–Sahlin_1 model and/or the Peppas–Sahlin_1 with *T_lag_* model are not useful for the interpretation of drug release mechanisms, although if the model fit is very good, such models can still be useful for other purposes (general representation of a drug release profile, the assessment of the fraction of drug released in chosen time points, in which experimental data were not obtained, etc.)

Carvedilol release from both PEG formulations and from the Kollidon^®^ 25 formulation seems to follow anomalous (non-Fickian) transport, whereas for the PEO formulation and the Kollidon^®^ 90 F formulation, evidence points to Super Case–II transport (see Table 2 and Table 3). In the first case, carvedilol release is therefore governed by diffusion and swelling, where rates of both are comparable. This indicates that the time-dependent anomalous effects are caused simultaneously by the diffusion process and by the slow rearrangement of polymeric chains. In the second case, an extreme form of transport seems to be taking place, characterised by tension and breaking of the polymer during the sorption process, also sometimes called ‘solvent crazing’. This is evident from the exponential shape of the carvedilol release profile in the initial stage of carvedilol release, where the beginning exponential-shaped part of the carvedilol release profile is just the initial part of the whole sigmoid-shaped carvedilol release profile observed in the PEO and the Kollidon^®^ 90 F formulations (see model fitting summary for the PEO formulation and the Kollidon^®^ 90 F formulation) [1,2,15,48,49]. 

In the mannitol group, both crystalline mannitol formulations, the C*Pharm Mannidex 16700 formulation and the PEARLITOL^®^ 160C formulation, seem to release carvedilol differently. For the C*Pharm Mannidex 16700 formulation, data point to anomalous (non-Fickian) transport (see Table 2 and Table 3), whereas for the PEARLITOL^®^ 160C formulation, the results are not consistent for all four tested tablets (see the model fitting summary for the PEARLITOL^®^ 160C formulation for models mentioned in Table 3). Carvedilol release from three of the four tablets seems to be consistent with anomalous (non-Fickian) transport, while in the case of one of the tablets, the Korsmeyer–Peppas model points to Case–II transport, meaning carvedilol is released by the swelling or relaxation of polymeric chains; the Korsmeyer–Peppas with F_0_ model even points to Super Case–II transport. This demonstrates that perhaps in some formulations demonstrating larger intertablet carvedilol release variability, the mechanism of carvedilol release could in some cases differ even among individual tablets. For both spray-dried mannitol formulations, the Parteck^®^ M 100 and the Parteck^®^ M 200 formulation, data point to carvedilol release by the Super Case–II transport mechanism (see Table 2 and Table 3). In the lactose group, the carvedilol release from the Lactochem^®^ Crystals, the SuperTab^®^ 11SD, and the Tablettose^®^ 70 formulations are consistent with the Fickian diffusion mechanism (see Table 2 and Table 3), meaning that the dissolution media transport rate or diffusion is much greater than the process of polymeric chain relaxation. These are the three lactose formulations exhibiting significant burst release (see Appendix A). In the case of the Lactochem^®^ Fine Powder and the FlowLac^®^ 100 formulations, evidence points to anomalous (non-Fickian) transport (see Table 2 and Table 3). Previous research does not provide a clear rule explaining the observed groupings for lactose formulations in terms of their assumed carvedilol release mechanism or the overall similarity of their carvedilol release profiles within each group [1,2,15,45,48,49,51].

For the sucrose, maltodextrin, DCP, MCC, and EC formulations, carvedilol release from tablets seems to follow anomalous (non-Fickian) transport (see Table 2, Table 3 and Table 4). For all these formulations, the Peppas–Sahlin_1 model and/or the Peppas–Sahlin_1 with *T_lag_* model produces a very good fit and points to different contributions of Fickian diffusion and Case–II polymeric relaxation. Interestingly, the Starch 1500^®^ formulations behave differently. For the Starch 1500^®^ formulation with ↓PS, data point to carvedilol being likely predominately released via the Fickian diffusion mechanism, although it is hard to be certain because the fit of none of the models used in carvedilol release mechanism analysis is particularly good. In the case of the Starch 1500^®^ formulation with ↑PS, the results are somewhat mixed. The Korsmeyer–Peppas with F_0_ model points to anomalous (non-Fickian) transport, although the fit of the model is again not particularly good. In contrast, the fit of the Peppas–Sahlin_1 model is very good and the model parameters show a large contribution of Fickian diffusion to the carvedilol release [1,2,15,16,48,49].

Overall, the results show the usefulness of the utilized mathematical models for model-dependent assessment of the carvedilol release mechanism. However, there is some uncertainty present in these assessments, as in some cases the fit of the models needed to be utilized for model-dependent assessment of the carvedilol release mechanism is not satisfactorily good. In these cases, the carvedilol release mechanism determination via the model-dependent approach remains somewhat elusive.

## 4. Conclusions

Mathematical models play a crucial role in analysing drug release from HPMC-based matrix tablets. Our research, conducted on a comprehensive collection of carvedilol controlled-release data, has yielded several conclusions.

No single model consistently provided a satisfactory fit, necessitating consideration of different models to accurately model carvedilol release. The mathematical structure of the models played a crucial role in their ability to fit the raw dissolution data, in addition to the number of parameters in each model. Surprisingly, the Makoid–Banakar model proved highly useful, despite generally receiving less recognition compared to the Higuchi, Korsmeyer–Peppas, and Peppas–Sahlin models in the field of controlled-release matrix tablets. Nevertheless, our results demonstrate that it is not suitable for all cases.

Applying the Higuchi, Korsmeyer–Peppas, and Peppas–Sahlin models to the relevant portion of dissolution data determined by the paired *t*-test has demonstrated that these models can sometimes accurately represent a larger portion of a drug’s controlled-release profile than the initial 60%, despite not being designed for this purpose.

Addressing the presence of a plateau in drug release within dissolution data is necessary, as it can hinder model fitting for certain models.

Comparing model-dependent estimations of burst release and lag time proved challenging when different mathematical models were employed or when model fit was inadequate or incomparable. Utilizing a more general fitting approach such as LOESS offers advantages by enabling the modelling of various release profile shapes using a unified approach. However, a high-resolution experimental dissolution data set is necessary for accurate modelling—especially when employing LOESS compared to the mathematical models used in this study. Furthermore, special attention should be paid to drug release profiles with sigmoid shapes, as our research demonstrated that models unable to capture this shape may inaccurately estimate lag time. Additionally, models unable to capture the initial curvature of drug release may overestimate burst release.

The RSS proved to be a suitable criterion for the overall comparison of model fit when the same number of experimental dissolution data points were used in fitting all models. However, our research has shown that visual examination of model fit and critical inspection of fitted model parameters, particularly those indicating burst release or lag time, are essential for selecting the most accurate models on an individual basis, alongside the utilization of general goodness of fit criteria. Additionally, reporting the range or standard deviation of goodness of fit criteria values along with the average value provides insights into each model’s ability to fit individual drug release profiles within the same formulation or batch.

Lastly, the model-dependent analysis of drug release mechanisms can be useful but proved challenging in cases where none of the models provided a very good fit. There is no universal guideline for determining the level of model fit quality necessary for valid model-dependent analysis of drug release mechanisms. Establishing practical guidelines in this regard is needed.

Our research provides comprehensive insights into the challenges associated with the practical application of mathematical models to controlled-release drug profiles. It critically evaluates the performance of a comprehensive set of models and illustrates key aspects of selecting the most accurate mathematical models. As such, it makes a significant contribution to the field of modelling controlled-release drug profiles.

Additional data generated in the present study, not included in the article publication and its supplements, are available from the authors.

## Figures and Tables

**Figure 1 pharmaceutics-16-00498-f001:**
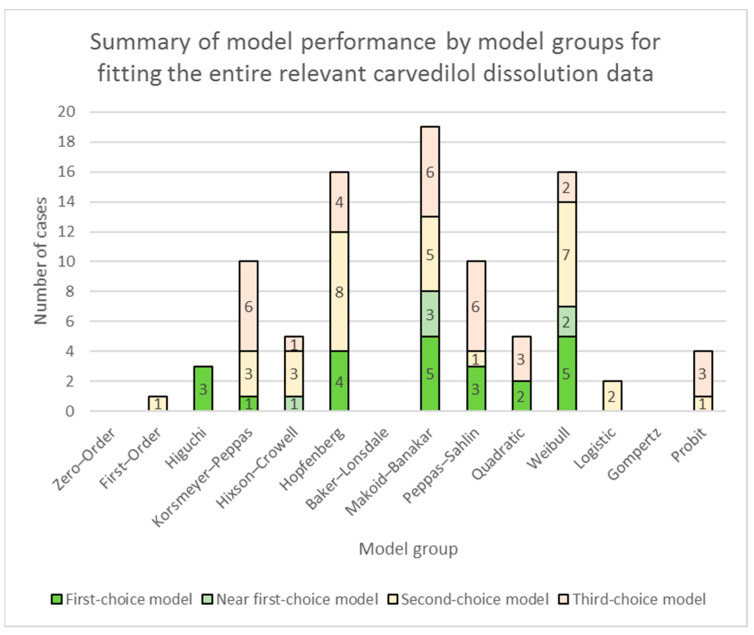
Summary of model performance by model groups (see Table 1) for fitting the entire relevant carvedilol dissolution data. The figure depicts the number of times one or more models in a model group performed as a first-choice model, a near first-choice model, a second-choice model, or a third-choice model. The height of each column depicts the total number of times one or more models from individual model groups were chosen as candidates for carvedilol release modelling.

**Table 1 pharmaceutics-16-00498-t001:** Mathematical model groups and models available in DDSolver for fitting to dissolution data [44].

Model Group	Model	Equation	Parameter(s)
Zero–order	Zero–order	$F=k_{0}\cdot t$	*k* _0_
	Zero–order with *T_lag_*	$F=k_{0}\cdot \left(t-T_{lag}\right)$	*k*_0_, *T_lag_*
	Zero–order with *F*_0_	$F={F_{0}+k}_{0}\cdot t$	*k*_0_, *F*_0_
First–order	First–order	$F=100\cdot \left(1-e^{-k_{1}\cdot t}\right)$	*k* _1_
	First–order with *T_lag_*	$F=100\cdot \left[1-e^{-k_{1}\cdot \left(t-T_{lag}\right)}\right]$	*k*_1_, *T_lag_*
	First–order with *F_max_*	$F=F_{max}\cdot \left(1-e^{-k_{1}\cdot t}\right)$	*k*_1_, *F_max_*
	First–order with *T_lag_* and *F_max_*	$F=F_{max}\cdot \left[1-e^{-k_{1}\cdot \left(t-T_{lag}\right)}\right]$	*k*_1_, *T_lag_*, *F_max_*
Higuchi	Higuchi	$F=k_{H}\cdot t^{0.5}$	*k_H_*
	Higuchi with *T_lag_*	$F=k_{H}\cdot {\left(t-T_{lag}\right)}^{0.5}$	*k_H_*, *T_lag_*
	Higuchi with *F*_0_	$F={F_{0}+k}_{H}\cdot t^{0.5}$	*k_H_*, *F*_0_
Korsmeyer–Peppas	Korsmeyer–Peppas	$F=k_{KP}\cdot t^{n}$	*k_KP_*, *n*
	Korsmeyer–Peppas with *T_lag_*	$F=k_{KP}\cdot {\left(t-T_{lag}\right)}^{n}$	*k_KP_*, *n*, *T_lag_*
	Korsmeyer–Peppas with *F*_0_	$F={F_{0}+k}_{KP}\cdot t^{n}$	*k_KP_*, *n*, *F*_0_
Hixson–Crowell	Hixson–Crowell	$F=100\cdot \left[1-{\left(1-k_{HC}\cdot t\right)}^{3}\right]$	*k_HC_*
	Hixson–Crowell with *T_lag_*	$F=100\cdot \left\{1-{\left[1-k_{HC}\cdot \left(t-T_{lag}\right)\right]}^{3}\right\}$	*k_HC_*, *T_lag_*
Hopfenberg	Hopfenberg	$F=100\cdot \left[1-{\left(1-k_{HB}\cdot t\right)}^{n}\right]$	*k_HB_*, *n*
	Hopfenberg with *T_lag_*	$F=100\cdot \left\{1-{\left[1-k_{HB}\cdot \left(t-T_{lag}\right)\right]}^{n}\right\}$	*k_HB_*, *n*, *T_lag_*
Baker–Lonsdale	Baker–Lonsdale	$\frac{3}{2}\cdot \left[1-{\left(1-\frac{F}{100}\right)}^{\frac{2}{3}}\right]-\frac{F}{100}=k_{BL}\cdot t$	*k_BL_*
	Baker–Lonsdale with *T_lag_*	$\frac{3}{2}\cdot \left[1-{\left(1-\frac{F}{100}\right)}^{\frac{2}{3}}\right]-\frac{F}{100}=k_{BL}\cdot \left(t-T_{lag}\right)$	*k_BL_*, *T_lag_*
Makoid–Banakar	Makoid–Banakar	$F=k_{MB}\cdot t^{n}\cdot e^{-k\cdot t}$	*k_MB_*, *n*, *k*
	Makoid–Banakar with *T_lag_*	$F=k_{MB}\cdot {\left(t-T_{lag}\right)}^{n}\cdot e^{-k\cdot \left(t-T_{lag}\right)}$	*k_MB_*, *n*, *k*, *T_lag_*
Peppas–Sahlin	Peppas–Sahlin_1	$F=k_{1}\cdot t^{m}+k_{2}\cdot t^{2m}$	*k*_1_, *k*_2_, *m*
	Peppas–Sahlin_1 with *T_lag_*	$F=k_{1}\cdot {\left(t-T_{lag}\right)}^{m}+k_{2}\cdot {\left(t-T_{lag}\right)}^{2m}$	*k*_1_, *k*_2_, *m*, *T_lag_*
	Peppas–Sahlin_2	$F=k_{1}\cdot t^{0.5}+k_{2}\cdot t$	*k*_1_, *k*_2_
	Peppas–Sahlin_2 with *T_lag_*	$F=k_{1}\cdot {\left(t-T_{lag}\right)}^{0.5}+k_{2}\cdot \left(t-T_{lag}\right)$	*k*_1_, *k*_2_, *T_lag_*
Quadratic	Quadratic	$F=100\cdot \left(k_{1}\cdot t^{2}+k_{2}\cdot t\right)$	*k*_1_, *k*_2_
	Quadratic with *T_lag_*	$F=100\cdot \left[k_{1}\cdot {\left(t-T_{lag}\right)}^{2}+k_{2}\cdot \left(t-T_{lag}\right)\right]$	*k*_1_, *k*_2_, *T_lag_*
Weibull	Weibull_1	$F=100\cdot \left[1-e^{-\frac{{\left(t-T_{i}\right)}^{\beta }}{\alpha }}\right]$	*α*, *β*, *T_i_*
	Weibull_2	$F=100\cdot \left(1-e^{-\frac{t^{\beta }}{\alpha }}\right)$	*α*, *β*
	Weibull_3	$F=F_{max}\cdot \left(1-e^{-\frac{t^{\beta }}{\alpha }}\right)$	*α*, *β*, *F_max_*
	Weibull_4	$F=F_{max}\cdot \left[1-e^{-\frac{{\left(t-T_{i}\right)}^{\beta }}{\alpha }}\right]$	*α*, *β*, *T_i_*, *F_max_*
Logistic	Logistic_1	$F=100\cdot \frac{e^{\alpha +\beta \cdot \log(t)}}{1+e^{\alpha +\beta \cdot \log(t)}}$	*α*, *β*
	Logistic_2	$F=F_{max}\cdot \frac{e^{\alpha +\beta \cdot \log(t)}}{1+e^{\alpha +\beta \cdot \log(t)}}$	*α*, *β*, *F_max_*
	Logistic_3	$F=F_{max}\cdot \frac{1}{1+e^{-k\cdot \left(t-\gamma \right)}}$	*k*, *γ*, *F_max_*
Gompertz	Gompertz_1	$F=100\cdot e^{-\alpha \cdot e^{-\beta \cdot log(t)}}$	*α*, *β*
	Gompertz_2	$F=F_{max}\cdot e^{-\alpha \cdot e^{-\beta \cdot log(t)}}$	*α*, *β*, *F_max_*
	Gompertz_3	$F=F_{max}\cdot e^{-e^{-k\cdot (t-\gamma )}}$	*k*, *γ*, *F_max_*
	Gompertz_4	$F=F_{max}\cdot e^{-{\beta \cdot e}^{-k\cdot t}}$	*k*, *β*, *F_max_*
Probit	Probit_1	$F=100\cdot \phi \left[\alpha +\beta \cdot log(t)\right]$	*α*, *β*
	Probit_2	$F=F_{max}\cdot \phi \left[\alpha +\beta \cdot log(t)\right]$	*α*, *β*, *F_max_*
*Explanation of symbols used*			
*F*	the fraction (%) of drug released in time *t*		
*F* _0_	the initial fraction of the drug in the solution resulting from a burst release		
*F_max_*	the maximum fraction of the drug released at infinite time		
*t*	time		
*T_lag_*, *T_i_*	the lag time prior to drug release or the location parameter		
*k* _0_	the zero–order release constant		
*k* _1_	the first–order release constant		
*k_H_*	the Higuchi release constant		
*k_KP_*, *n*	*k_KP_* is the release constant in the Korsmeyer–Peppas model and its variations (incorporating *T_lag_* or *F*_0_) incorporating structural and geometric characteristics of the DDS; *n* is the diffusional exponent in the Korsmeyer–Peppas model and its variations (incorporating *T_lag_* or *F*_0_) indicating the drug-release mechanism		
*k_HC_*	the release constant in the Hixson–Crowell model		
*k_HB_*, *n*	*k_BH_* is the combined constant in the Hopfenberg model, *k_HB_* = *k*_0_/(*C*_0_*∙a*_0_), where *k*_0_ is the erosion rate constant, *C*_0_ is the initial concentration of drug in the matrix, and *a*_0_ is the initial radius for a sphere or cylinder or the half thickness for a slab; *n* is 1, 2, and 3 for a slab, cylinder, and sphere, respectively		
*k_BL_*	*k_BL_* is the combined constant in the Baker–Lonsdale model, *k_BL_* = [3*∙D∙C_s_*/(*r*_0_^2^*∙C*_0_)], where *D* is the diffusion coefficient, *C_s_* is the saturation solubility, *r*_0_ is the initial radius for a sphere or cylinder or the half-thickness for a slab, and *C*_0_ is the initial drug loading in the matrix		
*k_MB_*, *n*, *k*	empirical parameters in the Makoid–Banakar model (*k_MB_*, *n*, *k* > 0)		
*k*_1_, *k*_2_, *m*	*k*_1_ is the constant related to Fickian kinetics; *k*_2_ is the constant related to Case–II relaxation kinetics; *m* is the diffusional exponent for a device of any geometric shape, which exhibits controlled release		
*k*_1_, *k*_2_	*k*_1_ is the constant denoting the relative contribution of *t*^0.5^-dependent drug diffusion to drug release; *k*_2_ is the constant denoting the relative contribution of *t*-dependent polymer relaxation to drug release		
*k*_1_, *k*_2_	*k*_1_ is the constant in the Quadratic model denoting the relative contribution of *t*^2^-dependent drug release; *k*_2_ is the constant in the Quadratic model denoting the relative contribution of *t*-dependent drug release		
*α*, *β*	*α* is the scale parameter; *β* is the shape parameter which characterises the curve as either exponential (*β* = 1; case 1), sigmoid, or S-shaped, with upward curvature followed by a turning point (*β* > 1; case 2), or parabolic, with a higher initial slope and after that consistent with the exponential (*β* < 1; case 3)		
*α*, *β*	*α* is the scale factor in the Logistic_1 and the Logitic_2 models; *β* is the shape factor in the Logistic_1 and Logistic_2 models		
*k*, *γ*	*k* is the shape factor in Logistic_3 model; *γ* is the time at which *F* = *F_max_*/2		
*α*, *β*	*α* is the scale factor in the Gompertz_1 and the Gompertz_2 models; *β* is the shape factor in the Gompertz_1 and the Gompertz_2 models		
*k*, *γ*	*k* is the shape factor in the Gompertz_3 model; *γ* is the time at which *F* = *F_max_*/*exp*(1) ≈ 0.368*∙F_max_*		
*β*, *k*	*β* is the scale factor in the Gompertz_4 model; *k* is the shape factor in the Gompertz_4 model		
*φ*, *α*, *β*	*Φ* is the standard normal distribution; *α* is the scale factor in the Probit models; *β* is the shape factor in the Probit models		

**Table 2 pharmaceutics-16-00498-t002:** Criteria for interpreting the carvedilol release mechanism from the value of the diffusional exponent *n* using the Korsmeyer–Peppas models.

Value of the Diffusional Exponent *n* in the Fitted Korsmeyer–Peppas Model	Drug Release Mechanism
0.45	Fickian diffusion
0.45 < *n* < 0.89	Anomalous (non-Fickian) transport
0.89	Case–II transport
>0.89	Super Case–II transport

**Table 3 pharmaceutics-16-00498-t003:** Summary of Higuchi and Korsmeyer–Peppas models’ data used for model-dependent analysis of carvedilol release mechanism. Results are presented as average ± one standard deviation. Models, which also performed as first-choice, near first-choice, second-choice, or third-choice models with respect to their fit to carvedilol dissolution data up to app. 60% of carvedilol released, are colour-coded in the same way as described in Section 2.2.5. In addition, data from models, which were considered important for the model-dependent analysis of the carvedilol mechanism, are bolded.

Formulation Id	Higuchi Model	Higuchi with *T_lag_* Model	Higuchi with *F*_0_ Model	Korsmeyer–Peppas Model		Korsmeyer–Peppas with *T_lag_* Model		Korsmeyer–Peppas with *F*_0_ Model	
RSS/Model Parameter	RSS	RSS	RSS	*n*	RSS	*n*	RSS	*n*	RSS
Polyglykol^®^ 4000 P (up to app. 60% of carv. rel.)	163.62 ± 70.65	22.15 ± 14.1	**0.76 ± 0.94**	**0.847 ± 0.032**	**6.74 ± 2.71**	**0.674 ± 0.038**	**0.3 ± 0.28**	1.123 ± 0.065	27.76 ± 16.1
Polyglykol^®^ 4000 P (up to app. 75% of carv. rel.)	238.6 ± 108.06	67.68 ± 61.51	**1.34 ± 1.14**	0.82 ± 0.045	24.04 ± 13.55	**0.67 ± 0.042**	**4.19 ± 3.1**	1.06 ± 0.057	80.54 ± 34.94
Polyglykol^®^ 8000 P (up to app. 60% of carv. rel.)	310.1 ± 17.94	168.32 ± 27.34	**1.6 ± 0.62**	1.039 ± 0.04	19.31 ± 3.06	**0.84 ± 0.034**	**3.6 ± 0.36**	1.394 ± 0.046	74.66 ± 4.25
Polyglykol^®^ 8000 P (up to app. 75% of carv. rel.)	454.71 ± 33.45	169.72 ± 26.2	**2.12 ± 0.68**	1.031 ± 0.04	76.58 ± 4.85	**0.833 ± 0.033**	**24.01 ± 2.24**	1.273 ± 0.042	191.99 ± 10.55
POLYOXᵀᴹ WSR N-80 (LEO NF Grade)	651.64 ± 84.72	135.36 ± 20.9	70.08 ± 28.6	1.312 ± 0.081	68.09 ± 45.18	**1.161 ± 0.074**	**19.61 ± 14.96**	1.552 ± 0.084	230.05 ± 65.77
KOLLIDON^®^ 25	255.46 ± 220.44	119.36 ± 119.57	15.82 ± 10.54	0.881 ± 0.153	22.09 ± 16.63	**0.71 ± 0.12**	**10.78 ± 10.2**	1.087 ± 0.15	47.46 ± 22.69
KOLLIDON^®^ 90 F	973.61 ± 152.58	259.65 ± 88.31	185.62 ± 91.57	1.15 ± 0.049	52.57 ± 74.56	**1.043 ± 0.035**	**34.01 ± 30.83**	1.267 ± 0.048	138.64 ± 177.25
C*Pharm Mannidex 16700	281.42 ± 360.46	163.81 ± 182.16	**104.06 ± 177.61**	0.729 ± 0.111	166.52 ± 322.14	**0.628 ± 0.083**	**132.25 ± 234.79**	0.91 ± 0.108	244.31 ± 466.55
PEARLITOL^®^ 160C	245.82 ± 373.3	198.16 ± 206.04	72.42 ± 125.69	**0.723 ± 0.111**	**45.24 ± 82.01**	0.6 ± 0.085	83.63 ± 138.26	**0.911 ± 0.123**	**24.47 ± 44.82**
Parteck^®^ M 100 (up to app. 60% of carv. rel.)	594.97 ± 444.85	122.22 ± 95.88	**63.9 ± 64.02**	1.547 ± 0.227	98.83 ± 167.67	**1.223 ± 0.157**	**66.55 ± 93.53**	1.867 ± 0.23	186.5 ± 285.77
Parteck^®^ M 100 (up to app. 70% of carv. rel.)	900.52 ± 425.2	169.09 ± 84.22	**96.98 ± 57.96**	1.433 ± 0.081	236.99 ± 284.79	**1.177 ± 0.103**	**208.55 ± 313.26**	1.717 ± 0.128	576.12 ± 745.48
Parteck^®^ M 200	599.68 ± 411.3	92.75 ± 34.94	**45.9 ± 28.59**	1.213 ± 0.102	116.44 ± 206.34	**1.017 ± 0.101**	**66.78 ± 109.29**	1.492 ± 0.151	227.57 ± 377.98
Lactochem^®^ Crystals	205.22 ± 90.39	9.02 ± 4.1	**0.77 ± 0.87**	0.338 ± 0.029	5.88 ± 3.41	0.296 ± 0.026	19.42 ± 6.56	**0.453 ± 0.028**	**1.3 ± 0.99**
Lactochem^®^ Fine Powder	87.03 ± 38.61	136.13 ± 38.16	17.83 ± 11.03	**0.594 ± 0.036**	**7.29 ± 5.38**	0.536 ± 0.033	29.58 ± 14.66	**0.702 ± 0.034**	**7 ± 10.3**
SuperTab^®^ 11SD	598.31 ± 326.66	7.26 ± 4.39	**0.67 ± 0.39**	0.247 ± 0.034	10.38 ± 3.17	0.213 ± 0.029	23.73 ± 5.28	**0.35 ± 0.047**	**4.36 ± 1.81**
FlowLac^®^ 100	78.94 ± 37.09	213.3 ± 53.18	36.86 ± 5.77	0.538 ± 0.044	38.56 ± 9.48	0.487 ± 0.04	86.4 ± 12.6	**0.667 ± 0.03**	**5.34 ± 1.78**
Tablettose^®^ 70	406.43 ± 142.2	14.62 ± 6.65	**1.67 ± 1.92**	0.284 ± 0.032	3.76 ± 3.14	0.247 ± 0.028	13.96 ± 6.3	**0.391 ± 0.037**	**0.92 ± 0.7**
Granulated sugar N°1 600	388.67 ± 26.08	220.83 ± 17.64	68.92 ± 6.67	**0.762 ± 0.015**	**2.16 ± 1.29**	0.691 ± 0.017	25.34 ± 6.81	0.853 ± 0.02	15.68 ± 6.28
GLUCIDEX^®^ 19	454.02 ± 157.31	312.58 ± 45.09	133.11 ± 48.18	0.737 ± 0.058	41.49 ± 10.47	0.673 ± 0.049	98.75 ± 13.98	**0.84 ± 0.055**	**10.6 ± 11.42**
DI-CAFOS^®^ A12	320.53 ± 77.03	306.29 ± 63.52	84.67 ± 22.01	0.659 ± 0.011	22.12 ± 6.29	0.608 ± 0.01	75.42 ± 16.14	**0.75 ± 0.011**	**0.5 ± 0.27**
EMCOMPRESS^®^ Anhydrous	62.28 ± 41.64	241.08 ± 116.1	39.04 ± 21.61	0.508 ± 0.014	57.19 ± 27.15	0.468 ± 0.012	115.18 ± 46.28	**0.601 ± 0.015**	**12.73 ± 9.68**
AVICEL^®^ PH-102	517.13 ± 54.46	304.91 ± 44.76	127.2 ± 6.52	**0.73 ± 0.047**	**12.39 ± 15.85**	0.688 ± 0.051	39.37 ± 42.59	0.802 ± 0.05	29.69 ± 28.12
AVICEL^®^ PH-200	411.99 ± 106.09	259.37 ± 35.91	86.38 ± 31.85	**0.731 ± 0.026**	**2.98 ± 2**	0.675 ± 0.019	23.52 ± 21.65	0.795 ± 0.025	26.54 ± 18.42
ETHOCELᵀᴹ Standard 20 Premium	559.11 ± 101.29	257.92 ± 17.08	117.19 ± 28.67	**0.767 ± 0.03**	**3.28 ± 1.46**	0.715 ± 0.016	20.78 ± 10.09	0.834 ± 0.019	24.69 ± 18.73
STARCH 1500^®^ sample with ↓PS	90.31 ± 76.42	149.22 ± 102.66	**49.54 ± 16.88**	0.415 ± 0.043	149.36 ± 52.78	0.385 ± 0.04	222.83 ± 64.84	**0.505 ± 0.046**	**76.41 ± 38.97**
STARCH 1500^®^ sample with ↑PS	167.48 ± 71.61	419.18 ± 82.55	100.16 ± 34.97	0.52 ± 0.02	142.42 ± 37.83	0.485 ± 0.019	233.67 ± 51.87	**0.607 ± 0.02**	**55.47 ± 23.08**

**Table 4 pharmaceutics-16-00498-t004:** Summary of Higuchi and Peppas–Sahlin_1 and Peppas–Sahlin_1 with T_lag_ models’ data used for the model-dependent analysis of carvedilol release mechanism. Results are presented as average ± one standard deviation. Models, which also performed as first-choice, near first-choice, second-choice, or third-choice models with respect to their fit to carvedilol dissolution data up to app. 60% of carvedilol released, are colour-coded in the same way as described in Section 2.2.5. In addition, data from models, which were considered important for the model-dependent analysis of the carvedilol mechanism, are bolded.

Formulation Id	Peppas–Sahlin_1 Model			Peppas–Sahlin_1 with *T_lag_* Model		
RSS/Model Parameter	*k* _1_	*k* _2_	RSS	*k* _1_	*k* _2_	RSS
Polyglykol^®^ 4000 P (up to app. 60% of carv. rel.)	**1.378 ± 0.96**	**1.849 ± 0.383**	**5.07 ± 2.17**	/	/	/
Polyglykol^®^ 4000 P (up to app. 75% of carv. rel.)	2.717 ± 0.736	1.528 ± 0.333	17.15 ± 6.32	**5.757 ± 0.5**	**1.124 ± 0.299**	**6.25 ± 3.34**
Polyglykol^®^ 8000 P (up to app. 60% of carv. rel.)	−2.008 ± 0.559	2.482 ± 0.065	7.68 ± 1.04	/	/	/
Polyglykol^®^ 8000 P (up to app. 75% of carv. rel.)	−0.209 ± 0.655	2.05 ± 0.07	28.83 ± 4.07	**2.84 ± 0.742**	**1.666 ± 0.083**	**15.79 ± 2.47**
POLYOXᵀᴹ WSR N-80 (LEO NF Grade)	−2.429 ± 0.547	1.05 ± 0.062	4.54 ± 3.42	−1.692 ± 0.594	0.989 ± 0.068	4.17 ± 3.25
KOLLIDON^®^ 25	1.21 ± 2.823	1.353 ± 0.858	14.33 ± 13.01	**3.572 ± 2.494**	**1.046 ± 0.819**	**8.1 ± 7.87**
KOLLIDON^®^ 90 F	−1.19 ± 0.504	0.499 ± 0.036	18.71 ± 16.88	−0.928 ± 0.566	0.483 ± 0.035	18.55 ± 15.92
C*Pharm Mannidex 16700	**2.441 ± 1.025**	**0.548 ± 0.238**	**132.3 ± 258.2**	**3.476 ± 1.297**	**0.441 ± 0.209**	**117.35 ± 227.85**
PEARLITOL^®^ 160C	**1.08 ± 2.32**	**0.787 ± 0.549**	**19.8 ± 36.7**	2.228 ± 2.178	0.662 ± 0.547	25.39 ± 43.31
Parteck^®^ M 100 (up to app. 60% of carv. rel.)	−6.833 ± 3.787	3.049 ± 1.212	37.22 ± 49.09	/	/	/
Parteck^®^ M 100 (up to app. 70% of carv. rel.)	−5.072 ± 1.341	2.626 ± 0.573	101.13 ± 118.56	−2.53 ± 1.081	2.344 ± 0.509	94.82 ± 107.63
Parteck^®^ M 200	−3.645 ± 1.461	2.181 ± 0.678	35.28 ± 54.65	−1.407 ± 0.994	1.928 ± 0.601	32.36 ± 47.47
Lactochem^®^ Crystals	7.452 ± 0.539	−0.17 ± 0.059	16.77 ± 11.02	7.804 ± 0.768	−0.233 ± 0.057	49.14 ± 20.82
Lactochem^®^ Fine Powder	**2.233 ± 0.309**	**0.138 ± 0.023**	**2.17 ± 2.96**	2.487 ± 0.319	0.121 ± 0.023	3.27 ± 1.31
SuperTab^®^ 11SD	12.175 ± 3.512	−0.373 ± 0.108	43.64 ± 17	10.736 ± 1.736	−0.502 ± 0.143	127.08 ± 48.45
FlowLac^®^ 100	**2.369 ± 0.52**	**0.112 ± 0.028**	**5.71 ± 4.91**	2.584 ± 0.534	0.098 ± 0.029	14.95 ± 9.71
Tablettose^®^ 70	10.606 ± 1.644	−0.326 ± 0.083	21.9 ± 14.62	10.054 ± 0.81	−0.425 ± 0.081	74.42 ± 27.43
Granulated sugar N°1 600	**0.784 ± 0.099**	**0.189 ± 0.006**	**4.09 ± 0.39**	**0.941 ± 0.105**	**0.179 ± 0.006**	**2.77 ± 0.62**
GLUCIDEX^®^ 19	**0.541 ± 0.277**	**0.196 ± 0.027**	**8.64 ± 13.09**	**0.68 ± 0.278**	**0.188 ± 0.027**	**11.41 ± 12.9**
DI-CAFOS^®^ A12	**1.056 ± 0.039**	**0.11 ± 0.012**	**0.55 ± 0.37**	**1.148 ± 0.039**	**0.105 ± 0.011**	**1.17 ± 0.3**
EMCOMPRESS^®^ Anhydrous	**1.953 ± 0.052**	**0.058 ± 0.015**	**5.8 ± 2.63**	2.052 ± 0.053	0.052 ± 0.015	13.47 ± 4.34
AVICEL^®^ PH-102	**0.697 ± 0.149**	**0.094 ± 0.004**	**2.77 ± 1.62**	**0.756 ± 0.151**	**0.092 ± 0.004**	**1.96 ± 1.09**
AVICEL^®^ PH-200	**0.822 ± 0.158**	**0.1 ± 0.011**	**3.7 ± 2.14**	**0.895 ± 0.161**	**0.097 ± 0.011**	**1.54 ± 1.21**
ETHOCELᵀᴹ Standard 20 Premium	**0.609 ± 0.135**	**0.113 ± 0.009**	**7.09 ± 2.46**	**0.68 ± 0.137**	**0.11 ± 0.009**	**5.07 ± 1.94**
STARCH 1500^®^ sample with ↓PS	**2.267 ± 0.467**	**0.02 ± 0.017**	**54.07 ± 43.49**	2.337 ± 0.472	0.016 ± 0.017	76.64 ± 55.88
STARCH 1500^®^ sample with ↑PS	**1.427 ± 0.127**	**0.055 ± 0.01**	**15.35 ± 7.53**	1.484 ± 0.128	0.052 ± 0.01	23.68 ± 8.71

## Data Availability

The data presented in this study are available on request from the corresponding author.

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
