# Peer review of "Comparative Fitting of Mathematical Models to Carvedilol Release Profiles Obtained from Hypromellose Matrix Tablets"

_pharmaceutics, 2024, doi:10.3390/pharmaceutics16040498_

Round 1

Reviewer 1 Report

Comments and Suggestions for Authors

The authors present a detailed and extensive comparison of the fittings of many mathematical models with a large number of carvedilol release profiles (23 different systems X 4 independent measurements in each case). Despite the complexity of the project, the organization of the whole presentation (with the conveniently accessed supplementary material) is excellent.

There are two main issues to be addressed before publication:

(a) The number of fitting parameters is a very important element of a model. Models with a large number of parameters are not so interesting, as they can fit almost anything. This is the reasoning I guess of the FDA guidelines mentioned in the text, recommending models with no more than 3 parameters. It is not surprising that models with more parameters can in general provide better fits than models with less parameters (except for cases where that additional parameters are completely irrelevant). Therefore a comparison between models with different number of parameters does not provide any useful information about a potentially preferable model.

I strongly suggest to group the investigated mathematical models according to the number of their parameters. Then discuss separately each group, showing corresponding figures as the current ones, starting with the fewer number of parameters. This can reveal a suitable compromise between less parameters and acceptable fits.

(b) Some parameters in a model, like the lag time T_lag or the initial burst F_0,  have a clear physical meaning. For example negative values of T_lag or F_0 make no sense, regardless of the permission or not of such unphysical values by a software (like the used DDSolver). It may seem that a negative T_lag value in a model gives qualitatively similar burst release behavior like a positive F_0 and vis versa. However such parameter values are unacceptable on physical grounds. If one wants to describe burst release (or lag time) should use a positive F_0 (or T_lag). Even if these options are absent in DDSolver, they can be easily implemented in a simple homemade program.

Unphysical are also negative values of the k constants in the Peppas-Sahlin models.

Therefore, fittings with such unphysical parameter values should not be considered.

A few minor comments:

1) It is more appropriate to substitute the world "modelling" by "fitting" in several places throughout the manuscript, including the Title.

2) page 1, three lines before the end: "It is difficult to explain its usage in drug release kinetics in a theoretical way" (referring to the first-order model). This is not correct. The analytical solutions of diffusion equation have infinite sums of exponentials and a single exponential dominates at relatively long times (see for example doi: 10.1016/j.jconrel.2011.10.006 , doi: 10.1016/j.physa.2017.05.033 , doi: 10.3390/pr11123431 ).

3) Beginning of page 2: "the Weibull model ... do not have any rooting in the reality of drug release phenomena". This is not true for the Weibull model (see doi: 10.1140/epjst/e2016-02669-8 ).

4) page 7: "α is the scale parameter which defines the time scale of the process" (referring to the parameter α of the Weibull model). This parameter does not define any time scale because it does not have units of time. For a proper time scale parameter of the Weibull model see doi:10.1016/j.ijpharm.2008.09.051 

5) There are some mistakes in the discussion and the description of the Supplementary Material Tables given in the last paragraph of page 7 and the beginning of page 8. It should be corrected.

6) In page 17 it is mentioned that in some cases the Higuchi model with F_0 provides better fits than the corresponding Korsmeyer–Peppas model. This cannot be valid because the Higuchi model is a subcase of the Korsmeyer–Peppas model for n=0.5 . Therefore all the fitting results obtained by the Higuchi model can be also obtained by the  Korsmeyer–Peppas model for n=0.5 . The observed situtation should be a problem of inaccurate fitting process.

7) In the caption of Table 4: "Summary of Higuchi ...".

Results for Higuchi are not shown in this Table.

8) In the captions of both Table 3 and Table 4: "In addition, data from other models, which was considered important for the model–dependent analysis of the carvedilol mechanism, is bolded".

It is not clear in what "other models" is referred this sentence. It should be explained in more detail.

9) Regarding the model-dependent mechanisms discussed in section 3.3:

Additionally to the models considered in this part of the paper, it should be noted that the exponent of the Weibull model can be also used for mechanism interpretation (see  doi: 10.1016/j.ijpharm.2005.10.044  ).

However, the mechanism interpretation obtained from all these empirical models should be always considered with great caution. This may be the reason of the discrepancy noted at the beginning of page 22 ("This demonstrates ...")

10) Page 26 (Conclusions): the discussion in the 2nd and 3rd paragraph.

The Makoid-Banakar model does not receive much recognition because it exhibits qualitatively wrong behavior at relatively long times. It does not give a monotonously increasing release and the plateau at the end, both of which representing real features of a release profile. The same problem also holds for the quadratic potential for negative k_1.

Similar problems regarding the plateau have the Higuchi, Korsmeyer–Peppas and Peppas-Sahlin models.

This is the reason that all these models are not used as proper descriptions of the complete release profiles. Instead, only the first 60% of the release has been proposed for the latter family of models.

11) Page 31: Reference 30 is not complete.

Reviewer 2 Report

Comments and Suggestions for Authors

Ojsteršek et. al., submitted the paper entitled “Comparative modelling of carvedilol release profiles from hypromellose matrix tablets using mathematical models” to be published in “Pharmaceutics (I. F= 5.4)”. This is an impressive work, that can be published after minor revision.

1. Concise the introduction by justifying the importance of this modelling.

2. Enhance the explanation for equations tabulated in Table 1.

3. Results and sections must be more informative with added explanations for Tables 2 to 4.

4. Concise the conclusion section with merits of this paper.
